# Heparan sulfate promotes TRAIL-induced tumor cell apoptosis

Yin Luo[1], Huanmeng Hao[1], Zhangjie Wang[2], Chih Yean Ong[1], Robert Dutcher[3], Yongmei Xu[2], Jian Liu[2], Lars C Pedersen[4], Ding Xu[1]*

[1]Department of Oral Biology, School of Dental Medicine, University at Buffalo, The State University of New York, Buffalo, United States; [2]Division of Chemical Biology and Medicinal Chemistry, Eshelman School of Pharmacy, University of North Carolina, Chapel Hill, United States; [3]Macromolecular Structure Group, Epigenetics and Stem Cell Biology Laboratory, National Institute of Environmental Health Sciences, National Institutes of Health, Research Triangle Park, United States; [4]Genome Integrity and Structural Biology Laboratory, National Institute of Environmental Health Sciences, National Institutes of Health, Research Triangle Park, United States

*For correspondence:
dingxu@buffalo.edu

**Abstract** TRAIL (TNF-related apoptosis-inducing ligand) is a potent inducer of tumor cell apoptosis through TRAIL receptors. While it has been previously pursued as a potential anti-tumor therapy, the enthusiasm subsided due to unsuccessful clinical trials and the fact that many tumors are resistant to TRAIL. In this report, we identified heparan sulfate (HS) as an important regulator of TRAIL-induced apoptosis. TRAIL binds HS with high affinity ($K_D$ = 73 nM) and HS induces TRAIL to form higher-order oligomers. The HS-binding site of TRAIL is located at the N-terminus of soluble TRAIL, which includes three basic residues. Binding to cell surface HS plays an essential role in promoting the apoptotic activity of TRAIL in both breast cancer and myeloma cells, and this promoting effect can be blocked by heparin, which is commonly administered to cancer patients. We also quantified HS content in several lines of myeloma cells and found that the cell line showing the most resistance to TRAIL has the least expression of HS, which suggests that HS expression in tumor cells could play a role in regulating sensitivity towards TRAIL. We also discovered that death receptor 5 (DR5), TRAIL, and HS can form a ternary complex and that cell surface HS plays an active role in promoting TRAIL-induced cellular internalization of DR5. Combined, our study suggests that TRAIL-HS interactions could play multiple roles in regulating the apoptotic potency of TRAIL and might be an important point of consideration when designing future TRAIL-based anti-tumor therapy.

## eLife assessment

This **fundamental** study advances our understanding of TRAIL-induced apoptosis by defining how Heparan triggers this pathway at the molecular level. The evidence supporting the conclusions is **compelling**, with rigorous binding assays, structural methods, and cellular studies. The work will be of broad interest to cell biologists and biochemists.

## Introduction

The tumor-necrosis factor (TNF) superfamily consists of 19 protein ligands that have diverse biological functions mainly in the immune, nervous, and skeletal systems (*Vanamee and Faustman, 2018*). Interestingly, while most factors in the superfamily stimulate cell proliferation and differentiation, several members function as pro-apoptotic ligands (*Croft et al., 2013*). In fact, all known ligands that drive

extrinsic pathways of apoptosis, including FAS ligand (FASL), TRAIL, and TNF-related weak inducer of apoptosis, belong to the TNF superfamily. TNF superfamily ligands are expressed as type II transmembrane proteins and form homotrimers through their extracellular domains. These extracellular domains can be liberated from the cell surface by proteases and function as soluble cytokines.

TRAIL was originally identified as a highly potent pro-apoptotic factor for tumor cells (*Wiley et al., 1995*). The main physiological function of TRAIL is believed to be regulation of thymocyte apoptosis during negative selection (*Lamhamedi-Cherradi et al., 2003*). TRAIL is known to bind five members of the TNF receptor superfamily. Two of these, TRAIL-R1 (also known as DR4) and TRAIL-R2 (also known as DR5) are functional death receptors with intact intracellular death domains (*Pan et al., 1997*; *Walczak et al., 1997*). The other three receptors, including TRAIL-R3, TRAIL-R4, and osteoprotegerin, are decoy receptors because binding of TRAIL will not induce apoptosis due to a lack of functional death domain (*Degli-Esposti et al., 1997*; *Emery et al., 1998*; *Sheridan et al., 1997*). Binding of TRAIL to DR4 and DR5 triggers apoptotic cascade by recruiting Fas-associated protein with death domain (FADD), which in turn uses its death effector domains to recruit and activate pro-caspases-8 and –10 (*von Karstedt et al., 2017*).

TRAIL received plentiful attention due to its unique property of inducing apoptosis in many types of cancer cells without affecting healthy, non-transformed cells (*Ashkenazi et al., 1999*; *Chen et al., 2012a*; *Lemke et al., 2014*; *Voss et al., 2021*; *Walczak et al., 1999*). Studies suggest that TRAIL plays an important role in tumor immune surveillance by suppressing tumor growth and metastasis, and TRAIL expressed by natural killer T cells has been implicated to play a major role in this process (*Smyth et al., 2001*; *Takeda et al., 2002*). To harness the therapeutic potential of TRAIL-induced tumor cell apoptosis, two classes of therapeutics have been developed to stimulate TRAIL signaling in tumor cells, which include recombinant TRAIL and agonistic antibodies against TRAIL receptors (*Montinaro and Walczak, 2023*; *von Karstedt et al., 2017*). Although one recombinant form of TRAIL and multiple agonistic anti-TRAIL-R antibodies have advanced to clinical trials, they failed to provide clinical benefit to cancer patients (*Forero-Torres et al., 2013*; *Herbst et al., 2010*; *Kelley et al., 2001*; *Merchant et al., 2012*; *Papadopoulos et al., 2015*; *Paz-Ares et al., 2013*; *Soria et al., 2011*; *von Pawel et al., 2014*). Multiple factors have been attributed to the failure in clinical trials so far, which includes insufficient capacity of the agonists to induce higher-order clustering of TRAIL-Rs, the resistance of many primary tumors to TRAIL-R agonists monotherapy, and a lack of biomarkers to identify tumors that are most suitable for TRAIL-based therapy (*von Karstedt et al., 2017*). These roadblocks indicate that our understanding of TRAIL signaling system is far from complete, which clearly hampers realization of the full potential of TRAIL-based anti-tumor therapy.

Heparan sulfate (HS) is a linear, negatively charged polysaccharide found at the cell surface and in the extracellular matrix of all mammalian cells (*Bishop et al., 2007*; *Esko and Selleck, 2002*). HS performs its biological function by binding to hundreds of secreted and transmembrane proteins. HS regulates the functions of protein binding partners chiefly by inducing oligomerization, promoting protein-protein interactions and inducing conformational changes (*Xu and Esko, 2014*). In the current study, we discovered that TRAIL is a novel HS-binding protein and HS can regulate the biological function of TRAIL in a multifaceted manner. We found that cell surface HS proteoglycans are required for the full proapoptotic activity of TRAIL in multiple tumor cell lines, and the activity of TRAIL can be fully inhibited by soluble heparin. Mechanistically, we provided evidence that HS readily induces TRAIL to form higher order oligomers, and that HS is involved in TRAIL-induced TRAIL receptor internalization by mediating a stable complex with TRAIL and TRAIL receptors. Additionally, crystallization of murine TRAIL reveals that trimeric TRAIL can exist in a strand-swapped fashion, which has never been reported for a TNF superfamily member. These discoveries strongly suggest that HS plays an active role in TRAIL biology and HS-TRAIL interaction should be taken into consideration when developing future TRAIL-based anti-tumor therapy.

## Results
### TRAIL is a HS binding protein and binding involves three specific basic residues

To investigate whether TRAIL is an HS-binding protein, we first examined the binding of the recombinant, soluble, extracellular domain of TRAIL to heparin using heparin–Sepharose chromatography. We

found that both human TRAIL (hTRAIL) and murine TRAIL (mTRAIL) bind well to the heparin column, and the binding of mTRAIL (eluted by 775 mM NaCl) to the heparin-Sepharose column was stronger than that of hTRAIL (eluted by 560 mM NaCl) (*Figure 1A*). To investigate the HS–TRAIL interaction in greater detail, we performed surface plasmon resonance (SPR) analysis of the binding between hTRAIL and immobilized HS dodecasaccharide (HS 12mer, *Table 1*). Our results showed that hTRAIL binds HS 12mer with a $K_D$ of 73 nM (*Figure 1B*). To explicitly determine the HS-binding site of TRAIL, we performed site-directed mutagenesis of seven conserved lysine and arginine residues on mTRAIL, some of which (Arg119, Arg122, and Lys125) cluster (in triplicate) near the 'top' of the trimer and likely form the HS-binding site (*Figure 1C*). Binding of these mTRAIL mutants to heparin was evaluated by heparin–Sepharose chromatography. Among tested mutants, only three (R119A, R122A, and K125A) displayed a substantial reduction in binding to heparin (*Figure 1D*). Interestingly, these three residues are located at the N-terminus of the soluble TRAIL, which likely adopts a random coil structure because it was not visible in the crystal structure of soluble TRAIL (*Figure 1C*). Among the identified residues, Arg119 apparently makes the most significant contribution to HS-binding (*Figure 1E*). To confirm that human TRAIL utilizes a similar HS-binding site, we mutated Arg115 of human TRAIL (structurally homologous to murine Arg119) (*Cha et al., 2000*) to alanine and found indeed, human R115A also display a dramatic reduction in HS-binding (*Figure 1F*).

## HS induces TRAIL to form higher-order oligomers in a length-dependent manner

To understand the structural details of HS-TRAIL interaction, we examined complex-formation between mTRAIL and structural-defined HS oligosaccharides (6mer to 18mer, *Table 1*) by size exclusion chromatography (SEC). We found that when HS 6mer was incubated with mTRAIL, it failed to induce any visible shift in the elution position of TRAIL (retention volume 15.8 ml), and that unbound, excess 6mer could be found eluted at 18.6 ml (*Figure 2A*). When HS 8mer was incubated with mTRAIL, we observed a slight shift in the retention volume of TRAIL, which possibly indicates the complex formation between mTRAIL and 8mer. When larger HS oligosaccharides (10mer to 18mer) were incubated with mTRAIL, we observed they had progressively stronger effects on the retention position of TRAIL, clearly indicating complex formation between TRAIL and these larger HS oligosaccharides (*Figure 2A*). In this experiment we noticed that the retention position of the apo form of mTRAIL (predicted trimer MW = 60 kDa) was much later than the retention position of similar-sized proteins based on MW standards (*Figure 2A*). To confirm that our recombinant mTRAIL truly exists as a homotrimer, and to determine the oligomeric states of the complexes formed by mTRAIL and HS oligosaccharides, we performed multiangle light scattering (MALS) analysis of mTRAIL, mTRAIL/12mer complex and mTRAIL/18mer complex (*Figure 2B*). SEC-MALS analysis found that the apo form of mTRAIL has a MW of 63 kDa, which confirms that it is indeed a homotrimer. The MW of mTRAIL/12mer complex was found to be 72.8 kDa, which indicates that 12mer (MW = 3.2 kDa) could form a stable complex with mTRAIL homotrimer but could not induce mTRAIL to form a larger oligomer. In contrast, the MW of mTRAIL/18mer complex is significantly larger (102 kDa), which is close to the predicted MW of a mTRAIL hexamer (dimer of trimers, 120 kDa). This result suggests that longer HS oligosaccharides may induce mTRAIL to form larger oligomers. Indeed, we found that mTRAIL can form oligomers that appeared even bigger when incubated with low molecular weight heparin (LMWH, average 24mer) or full-length heparin (average 50mer) (*Figure 2C*). Of note, when we tried to perform similar experiments with hTRAIL, 12mer interaction caused rapid precipitation of hTRAIL, suggesting that HS also may induce hTRAIL oligomerization that is unstable in in vitro settings.

## Crystallization of mTRAIL

We determined the crystal structure of soluble mTRAIL to better compare to hTRAIL and understand the biophysical characterization of HS and mTRAIL interactions (*Table 2*). Though a HS 12mer was present in the crystallization conditions, the oligosaccharide was not visible in the electron density. Similar to hTRAIL, mTRAIL exists as a trimer composed of three β-scaffold core protomers with one protomer comprising the asymmetric unit (*Figure 3A and B*). A zinc ion, critical for activity (*Hymowitz et al., 2000*), is located on the threefold axis bound by cysteine 240 from each protomer and a chloride ion (*Figure 3B*). A major difference between TRAIL and other TNF family members is an insertion of ~12–16 residues between β-strands A and A″ known as the A-A″ loop (or AA″ loop, *Figure 3A*; *Cha*

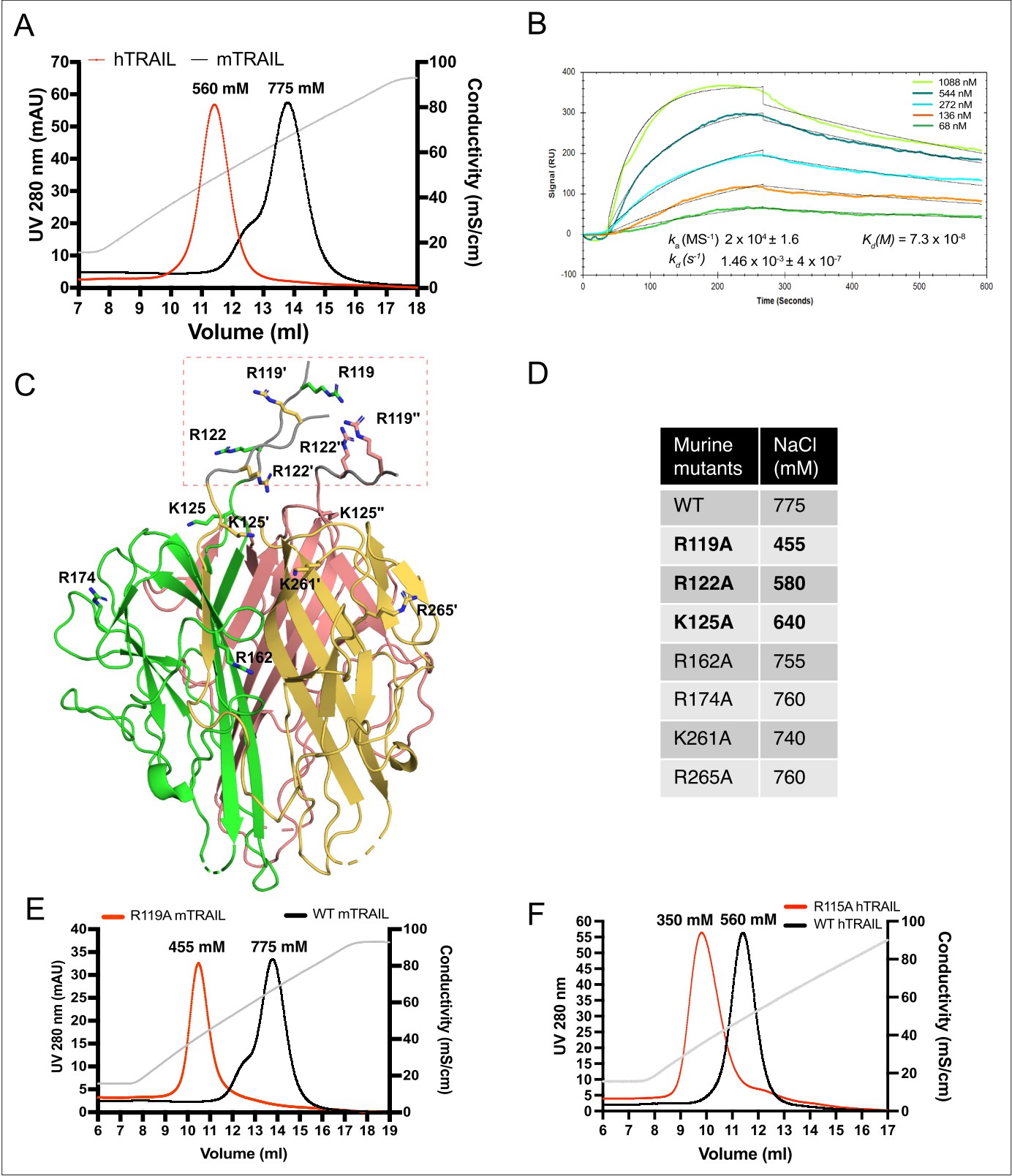

**Figure 1.** TNF-related apoptosis-inducing ligand (TRAIL) is a heparan sulfate (HS) binding protein and binding involves three N-terminal basic residues. (**A**) Binding of murine TRAIL (mTRAIL) and human TRAIL (hTRAIL) to heparin-Sepharose column. The gray line represents the salt gradient (in conductivity mS/cm, from 150 mM to 1 M). (**B**) Surface plasmon resonance (SPR) analysis of binding between hTRAIL and HS GlcNS6S- GlcA-GlcNS6S-(IdoA2S-GlcNS6S)4-GlcA dodecamer oligosaccharide. (**C**) Residues potentially involved in HS- binding. Crystal structure of mTRAIL homotrimer is shown

*Figure 1 continued on next page*

*Figure 1 continued*

in the cartoon. The three monomers are displayed in green, salmon, and gold, respectively. Because residues 118–123 were disordered in our mTRAIL structure, these residues (118PRGGRP123, backbone shown in gray random coils, enclosed in the red dashed rectangle) were manually modeled onto the last visible N-terminal residue (Q124) of the crystal structure of mTRAIL to allow displaying R119 and R122. (**D**) Salt elution position of wild-type (WT) or mutants mTRAIL on HiTrap heparin- Sepharose column. (**E**) Chromatogram of WT and R119A mTRAIL binding to heparin column. (**F**) Chromatogram of WT and R115A hTRAIL binding to heparin column.

*et al., 1999*). The N-terminal β-strand A of mTRAIL is found sandwiched between β-stands H and A″ of the neighboring protomer's central β-scaffold core, a pattern repeated for each protomer (*Figure 3A and B*). Clear electron density can be traced between this strand and β-strand A″ confirming the swapping of the N-terminal strand between the protomers. Interestingly, this swap has not been reported in other structures of TRAIL or TNF family members. In most structures of TRAIL reported some component of this loop is disordered (*Cha et al., 1999*; *Cha et al., 2000*; *Hymowitz et al., 1999*; *Hymowitz et al., 2000*). When we overlay the crystal structure of hTRAIL (PDB_ID:1DU3, co-crystal structure with DR5) with our structure of mTRAIL, we found that in the 1DU3 coordinates (*Cha et al., 2000*), the A-A″ loop is disordered between residues T135 and A146, however, the distance (36 Å) between them is likely too great for eleven amino acids to span (*Figure 3C*). Yet, T135 is located ~8 Å from A146 of the neighboring protomer, suggesting a strand swap might also be possible in hTRAIL.

## Small angle X-ray scattering (SAXS) analysis of TRAIL/oligosaccharide complexes

To gain better structural insights into HS-induced TRAIL oligomerization, we performed SAXS analysis of mTRAIL alone and mTRAIL in complex with either a 12mer or 18mer. In these experiments, the complexes were first resolved by size exclusion chromatography and the SAXS data were collected from the eluent in-line in real-time (*Figure 4—source data 1*). From the P(r) (paired distribution) plot (*Figure 4A*), it is obvious that compared to TRAIL alone, mTRAIL/18mer complex has substantially increased radius of gyration (Rg), and dramatically increased maximum dimension ($D_{max}$) (*Figure 4A*), indicating that mTRAIL/18mer complex likely adopts a larger oligomeric state. Consistent with the MW estimation of SEC-MALS analysis (*Figure 2B*), MW estimation based on SAXS data showed that the predicted MW of the mTRAIL/18mer complex almost doubles the predicted MW of TRAIL alone (105.5 vs 53.7 kDa), which strongly suggests that the TRAIL/18mer complex is likely a stable hexamer (dimer of homotrimers). Also consistent with our SEC-MALS analysis, SAXS data indicate that the dimension of mTRAIL/12mer is much smaller than mTRAIL/18mer but larger than the apo form of mTRAIL, suggesting that mTRAIL and HS 12mer likely formed a stable complex but the binding did not alter the oligomeric state of mTRAIL.

Using the scattering data of the mTRAIL/18mer complex, which we predict to be a dimer of homotrimers, we generated an *ab initio* molecular envelope using GASBOR software, based on P2 symmetry (*Figure 4B*). The best-fit GASBOR model ($\chi 2$=0.93) adopt a dumbbell shape that can accommodate two TRAIL trimers. For perspective we have modeled this in *Figure 4B* based on the lattice contact between two trimers found in the mTRAIL crystal structure. However, the relative orientation of the two trimers in the SAXS model cannot be definitively determined. This is due to the fact the TRAIL

**Table 1.** List of heparan sulfate (HS) oligosaccharides used in the study.

| Oligosaccharides | Structure |
| --- | --- |
| 6mer | GlcNS6S-GlcA-GlcNS6S-IdoA2S-GlcNS6S-GlcA-pNP* |
| 8mer | GlcNS6S-GlcA-GlcNS6S-(IdoA2S-GlcNS6S)$_2$-GlcA-pNp |
| 10mer | GlcNS6S-GlcA-GlcNS6S-(IdoA2S-GlcNS6S)$_3$-GlcA-pNP |
| 12mer | GlcNS6S-GlcA-GlcNS6S-(IdoA2S-GlcNS6S)$_4$-GlcA-pNP |
| 14mer | GlcNS6S-GlcA-GlcNS6S-(IdoA2S-GlcNS6S)$_5$-GlcA-pNP |
| 18mer | GlcNS6S-GlcA-GlcNS6S-(IdoA2S-GlcNS6S)$_7$-GlcA-pNP |

*pNP = p-nitrophenol. pNP has an UV absorbance peak at 310 nm, which also has significant absorbance at UV 280 nm.

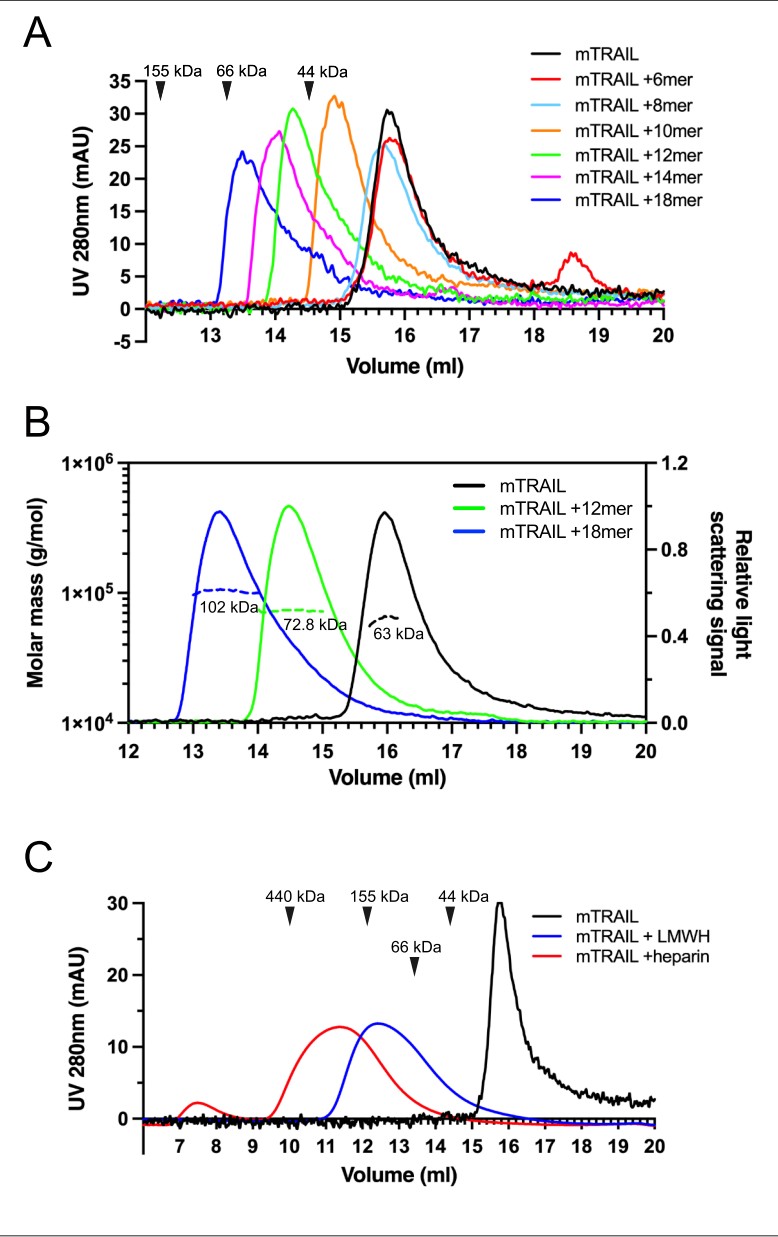

**Figure 2.** Heparan sulfate (HS) induces TNF-related apoptosis-inducing ligand (TRAIL) to form higher-order oligomers in a length-dependent manner. (**A**) Size exclusion chromatography (SEC) analysis of murine TRAIL (mTRAIL) in complex with HS oligosaccharides of different sizes (6mer, 8mer, 10mer, 12mer, 14mer, and 18mer) on Superdex200 Increase column. Elution position of the molecular weight standards (IgG, 155 kDa; BSA, 66 kDa, and ovalbumin, 44 kDa) are indicated with black triangles. (**B**) MW determination of TRAIL, TRAIL/12mer complex, and TRAIL/18mer complex by SEC-MALS. The MW data was plotted as dotted lines (left Y-axis) and the relative light scatter signals were plotted as solid lines (right Y-axis) (**C**) SEC analysis of mTRAIL in complex with low molecular weight heparin (LMWH) and full-length heparin.

trimer has similar dimensions vertically and horizontally (55–60 Å), which makes it possible for many different relative orientations to fit reasonably well into the GASBOR model.

## HS contributes to cell surface binding of TRAIL

Next, we sought to determine to what extent HS is involved in the binding of soluble TRAIL to breast cancer cells by using a flow-cytometry-based cell surface binding assay. As shown in *Figure 5A*, mTRAIL bound to the cell surface of murine breast cancer 4T1 cells in a dose-dependent manner.

**Table 2.** Data collection and refinement statistics.

| | mTRAIL[*,†] |
|---|---|
| PDB ID code | 8SLR |
| **Data collection** | |
| Space group | P4₁32 |
| Cell dimensions | |
| $a$, $b$, $c$ (Å) | 147.35, 147.35, 147.35 |
| α, β, γ (°) | 90, 90, 90 |
| Resolution (Å) | 50.00–2.40 (2.44-2.40)[‡] |
| $R_{sym}$ (%) | 11.0 (93.0) |
| $I / \sigma I$ | 4.6 (1.8) |
| Completeness (%) | 99.9 (100.0) |
| Redundancy | 9.4 (9.9) |
| | |
| **Refinement** | |
| Resolution (Å) | 40.50 (2.40) |
| No. reflections | 21,811 |
| $R_{work} / R_{free}$ (%) | 15.48/18.06 |
| No. atoms | |
| Protein | 1314 |
| Water | 105 |
| Other | 15 |
| *B*-factors | |
| Protein | 46.57 |
| Water | 51.61 |
| Other | 86.98 |
| R.m.s. deviations | |
| Bond lengths (Å) | 0.006 |
| Bond angles (°) | 0.817 |
| Ramachandran Plot | |
| Allowed (%) | 3.31 |
| Favored (%) | 96.7 |

[*]A single crystal was used to collect each dataset.

[†]These crystals were collected on the Southeast Regional Collaborative Access Team (SER-CAT) 22-ID beamline at the Advanced Photon Source at Argonne National Laboratory.

[‡]Values in parentheses are for the highest-resolution shell.

After removing cell surface HS with heparin lyase III (HL-III), we found binding of mTRAIL was reduced by 55% at 300 ng/ml (relative fluorescence units (RFU)=90 vs 40 in untreated and HL-III cells), and by 70% at 100 ng/ml (RFU reduced from 65 to 20). Similar HS-dependent binding was also observed in hTRAIL binding to human breast cancer MDA-MB-453 cells, which after HL-III treatment displayed 60% reduction in binding at 300 ng/ml and 67% reduction at 100 ng/ml (*Figure 5B*). Using the same assay, we also examined the binding of hTRAIL R115A mutant to MDA-MB-453 cells. Consistent with its greatly reduced binding to heparin-Sepharose (*Figure 1F*), R115A displayed 60% reduction in

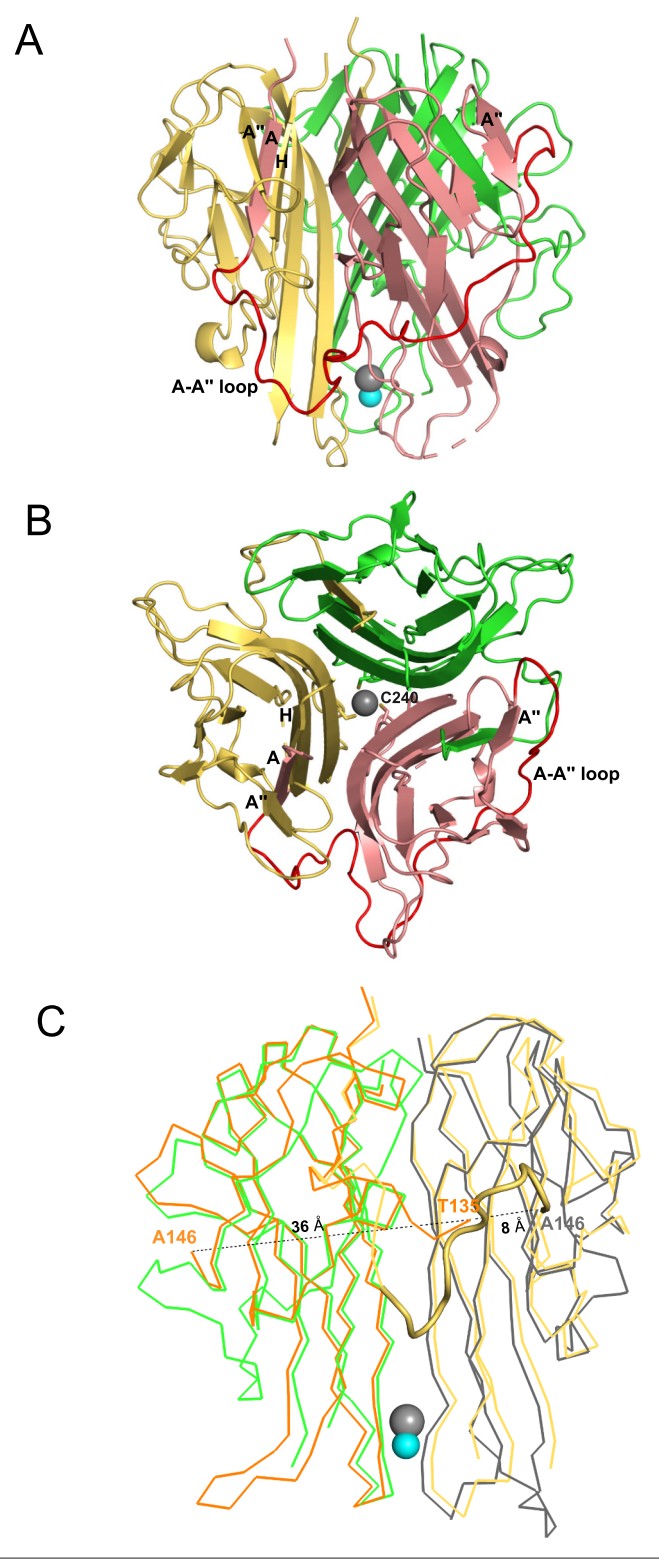

**Figure 3.** Crystal structural murine TRAIL (mTRAIL). (**A**) mTRAIL is observed as a strand-swapped homotrimer. Note the N-terminus strand A of one monomer (salmon) is inserted into the β-sheet of the neighboring monomer (yellow). Beta strands A and A" of the salmon monomer, and the strands A" and H of the yellow monomers are labeled. The loop connecting strands A and A" (The A-A" loop) of the salmon monomer is shown in red. Zinc and chloride are shown in cyan and gray spheres, respectively. (**B**) Looking down from the threefold axis of domain-

*Figure 3 continued on next page*

*Figure 3 continued*

swapped mTRAIL homotrimer. Side chains of Cys240, which are responsible for chelating Zn2+, are shown in sticks. (**C**) Overlay of the crystal structure of human TRAIL (hTRAIL) (1DU3) and mTRAIL. To facilitate visualization, only two monomers of hTRAIL (orange and gray) and two monomers of mTRAIL (green and yellow) are shown in the ribbon representation. In the hTRAIL structure, the fragment between T135 and A146 is missing, whereas the homologous fragment in mTRAIL structure is visible (yellow cartoon representation). Note the T135 of the orange human monomer is 36 Å away from A146, where it is only 8 Å from A146 of the neighboring gray human monomer.

binding to cell surface compared to WT hTRAIL (RFU reduced from 10 to 4) (*Figure 5C*). Of note, after cells are treated with HL-III, there is no difference in binding between WT and R115A hTRAIL, suggesting the R115A mutant fully retains its HS-independent binding capability to cell surface TRAIL receptors but completely lacks HS-dependent interaction with the cell surface (*Figure 5C*).

## Cell surface HS promotes TRAIL-induced breast cancer cell apoptosis

Since HS contributes to the binding of TRAIL to tumor cell surfaces, we wondered if HS plays a role in TRAIL-induced tumor cell apoptosis. We first tested this on the adherent breast cancer cell line MDA-MB-453 cells using Annexin V-FITC apoptosis assays. At 30 ng/ml, which we found is sufficient to induce maximum apoptosis in these cells, hTRAIL alone could induce around 8% of cells to undergo apoptosis (*Figure 6A and B*, *Figure 6—source data 1*). However, when cell surface HS was first removed by heparin lyase III (HL-III), TRAIL failed to induce apoptosis above the background level (*Figure 6A and B*). This result suggests that cell surface HS is essential for TRAIL-induced apoptosis in MDA-MB-453 cells. We further tested the impact of the addition of exogenous heparin in TRAIL-induced apoptosis. When heparin was added together with TRAIL to MDA-MB-453 cells, TRAIL-induced apoptosis was completely blocked (*Figure 6A and B*) presumably by competitively inhibiting TRAIL binding to HS on the cell surface. Combined, these results strongly suggest that cell surface HS can play a critical role in promoting the activity of TRAIL, and such a role is impaired when exogenous heparin is present, which can directly compete with cell surface HS for binding to TRAIL.

As an alternative method to examine apoptosis, we also performed TUNEL staining to examine the role of HS in TRAIL-induced apoptosis. Consistent with what we saw with Annexin V assay--there were

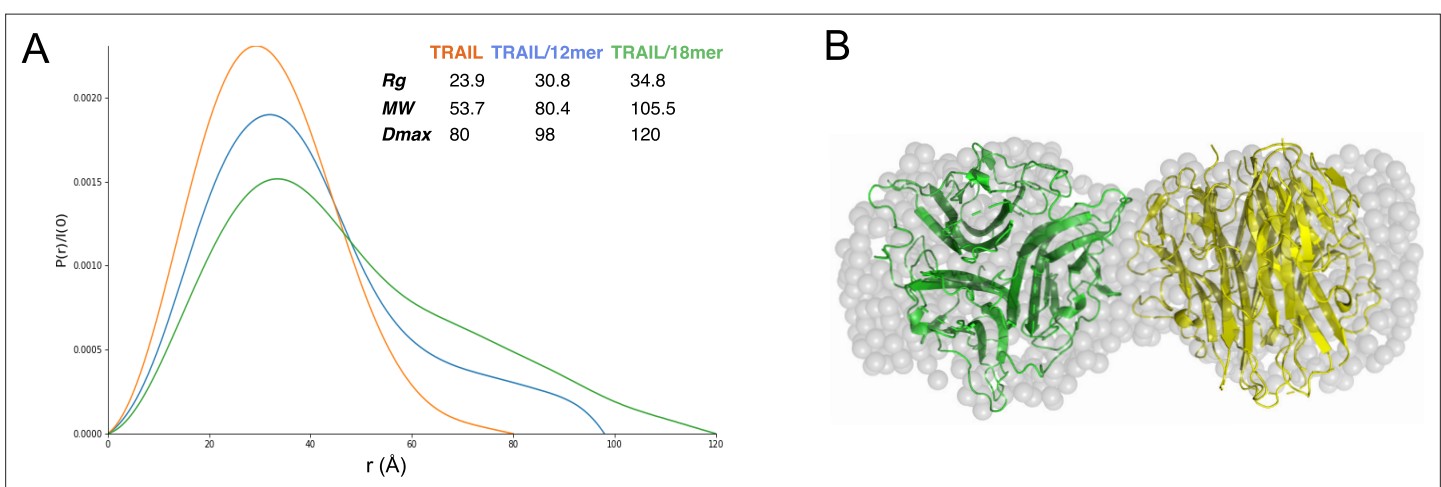

**Figure 4.** Structural analysis of TNF-related apoptosis-inducing ligand (TRAIL)/oligosaccharide complexes by small angle X-ray scattering (SAXS). (**A**) SAXS analysis of size exclusion chromatography (SEC)-purified TRAIL (red), TRAIL/12merNS2S6S complex (blue), and TRAIL/18merNS2S6S complex (green). Shown are overlays of P(r) function plots, along with Rg, Dmax, and MW values were determined from SAXS data. (**B**) GASBOR-generated ab initio model of TRAIL/18mer complex (with a $\chi$2=0.93) is shown in transparent gray beads. The crystallographic hexamer is manually superimposed onto the GASBOR model with one trimer green and the other yellow.

The online version of this article includes the following source data for figure 4:

**Source data 1.** Small angle X-ray scattering (SAXS) raw data file for apo TNF-related apoptosis-inducing ligand (TRAIL), TRAIL/12merNS2S6S complex, and TRAIL/18merNS2S6S complex.

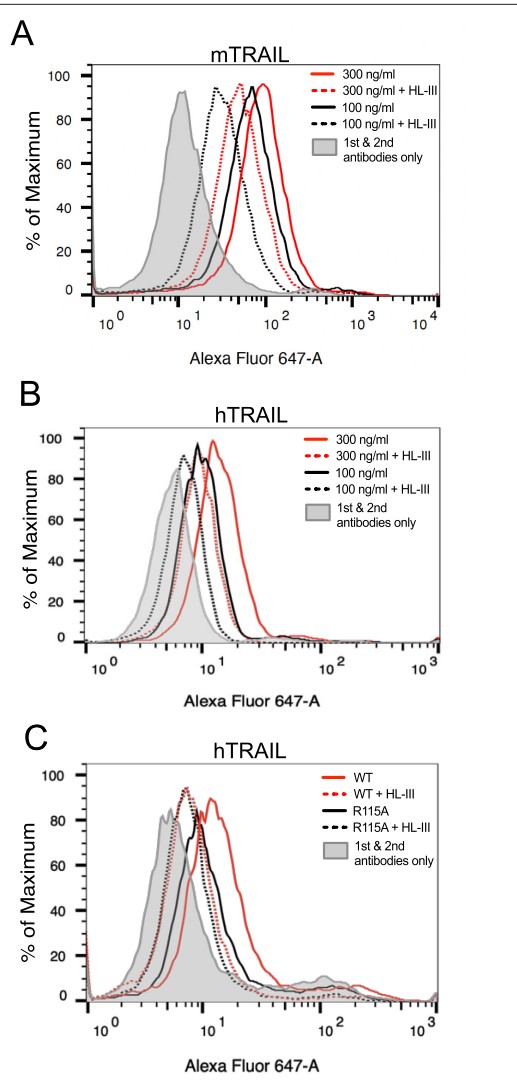

**Figure 5.** Heparan sulfate (HS) contributes to cell surface binding of TNF-related apoptosis-inducing ligand (TRAIL). Binding of mouse TRAIL (100 ng/ml and 300 ng/ml) to 4T1 breast cancer cells (**A**) and human TRAIL to MDA-MB-453 breast cancer cells (**B**), with or without heparin lyase III (HL-III) pretreatment, were determined by a FACS-based binding assay. The bound TRAIL were detected by staining with a goat anti-TRAIL antibody, followed by anti-goat-IgG Alexa-647. The shaded histogram is from cells stained only with primary and secondary antibodies. (**C**) Binding of wild-type (WT) and R115A hTRAIL (300 ng/ml) to MDA-MB-453 cells, with or without HL-III pretreatment were determined by a FACS-based binding assay.

many apoptotic cells in the TRAIL alone group, while very few apoptotic cells could be seen in the HL-III pretreated or heparin/TRAIL groups (**Figure 6C and D**, **Figure 6—source data 2**).

## Cell surface HS promotes TRAIL-induced myeloma cell apoptosis

To investigate whether the dependence of TRAIL on cell surface HS also applies to other types of tumor cells, we examined RPMI-8226 cells, a widely used human myeloma cell line. As reported, RPMI-8226 cells are highly sensitive to TRAIL-induced apoptosis (**Mitsiades et al., 2001**), requiring only 1 ng/ml hTRAIL to induce around 25% early apoptosis (**Figure 7A and C**, **Figure 7—source data 1**). As expected, the removal of cell surface HS significantly reduced TRAIL-induced apoptosis in RPMI-8226 cells (**Figure 7B and C**). After subtracting the background level of apoptosis (6%), we found that removal of cell surface HS resulted in 55% reduction in apoptosis at 1 ng/ml TRAIL and 44% reduction in apoptosis at 3 ng/ml TRAIL (**Figure 7B**, **Figure 7—source data 2**). Next, we examined the effect of adding exogenous heparin and found that only 1 μg/ml heparin was able to abolish TRAIL-induced apoptosis (**Figure 7D**, **Figure 7—source data 3**). HS 12mer was also effective in inhibiting apoptosis, albeit to a lesser degree compared to heparin (**Figure 7D**). This is perhaps unsurprising, given that longer oligosaccharide chains often compete better for binding than shorter chains, and cell surface HS is longer than either heparin or short, synthesized compounds. Finally, we examined the pro-apoptotic capacity of our HS binding-deficient mutant R115A. R115A showed significantly lower induction of apoptosis compared with WT hTRAIL at both 20 ng/ml and 100 ng/ml (**Figure 7E**, **Figure 7—source data 4**). While only 20 ng/ml of WT hTRAIL is sufficient to induce 20% apoptosis, the mutant TRAIL requires 100 ng/ml to induce the same level of apoptosis, indicating a fivefold reduction in potency. This result also supports the hypothesis that interaction between cell surface HS and TRAIL greatly enhances TRAIL activity.

## Cell surface HS level contributes to the sensitivity of myeloma cells towards TRAIL

Studies indicate that the sensitivities of myeloma cell lines to TRAIL varies substantially (**Gómez-Benito et al., 2007**; **Mitsiades et al., 2001**). In order to understand whether HS plays a role in regulating the sensitivity of different myeloma cells to TRAIL, we compared the HS contents in three different myeloma cell lines. Compared to RPMI-8226 cells (requiring only 1 ng/ml of TRAIL to induce 25% apoptosis, **Figure 7A**), U266 cells are moderately sensitive to TRAIL, while IM-9 cells are highly

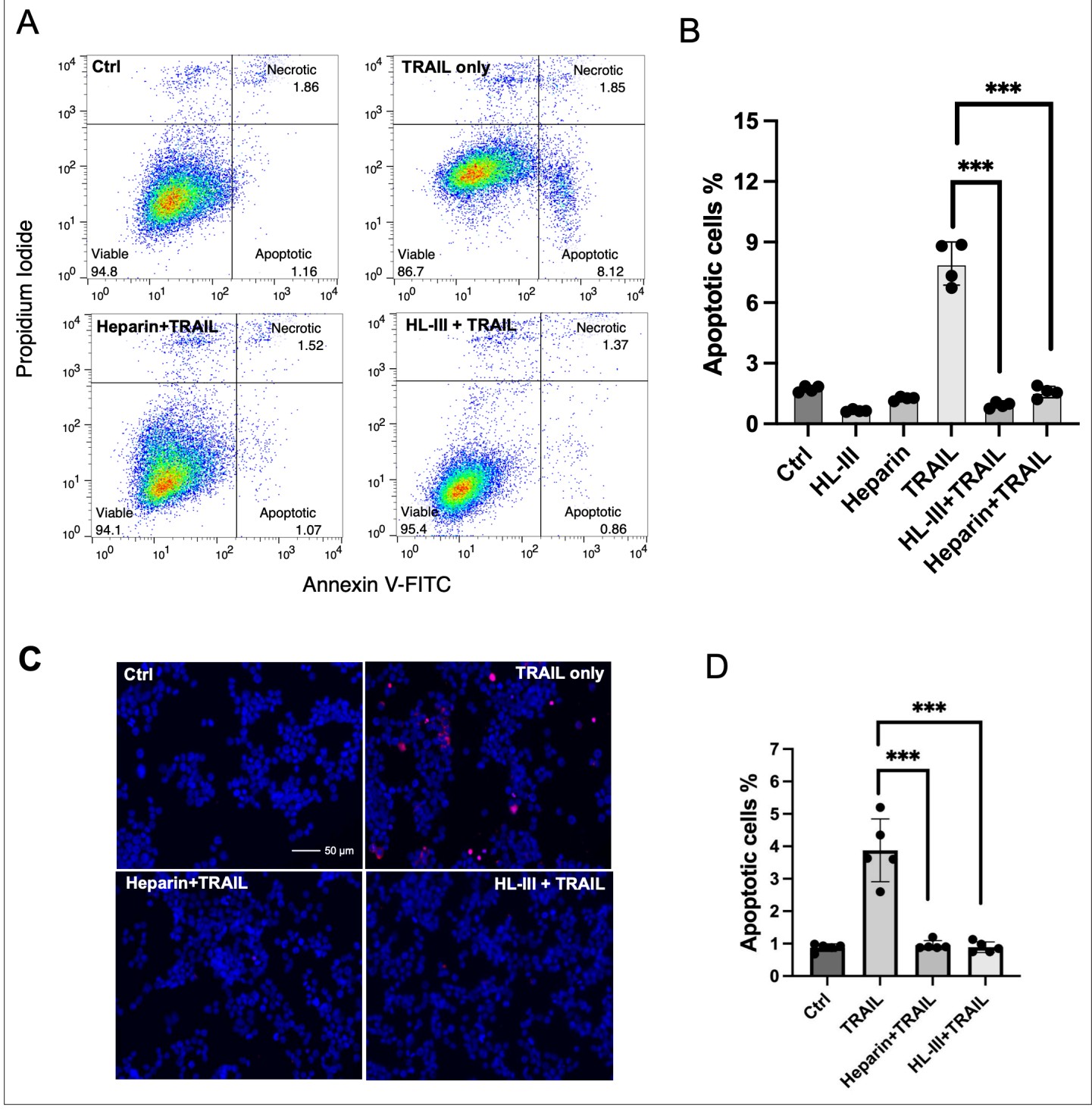

**Figure 6.** Cell surface heparan sulfate (HS) promotes TNF-related apoptosis-inducing ligand (TRAIL)-induced breast cancer cell apoptosis. (**A**) The representative Annexin V-FITC apoptotic assay plots and gating method of MDA-MB-453 cells treated with human TRAIL (hTRAIL) (30 ng/ml), in the presence or absence of heparin lyase III (HL-III) and exogenous heparin. Analysis was performed after cells were treated with hTRAIL for 6 hr. (**B**) Statistical analysis of TRAIL-induced early apoptotic cell population with various treatments. *** represents p<0.0001 by Student's t-test. Data are representative of at least three separate assays. (**C**) MDA-MB-453 cells were treated with TRAIL (30 ng/ml) in the presence or absence of HL-III and exogenous heparin for 6 hr. TUNEL staining was performed to visualize apoptosis. Apoptotic cells were stained red. Nuclei were stained blue with DAPI. Scale bar: 50 µm. (**D**) Statistical analysis of TUNEL staining.

The online version of this article includes the following source data for figure 6:

*Figure 6 continued on next page*

*Figure 6 continued*

**Source data 1.** Excel file with raw data used to generate *Figure 6B*.

**Source data 2.** Excel file with raw data used to generate *Figure 6D*.

resistant to TRAIL (*Figure 8A*, *Figure 8—source data 1*). Despite the difference in sensitivity, removal of cell surface HS reduced 50–60% of TRAIL-induced apoptosis in both IM-9 and U266 cells, suggesting cell surface HS promotes TRAIL-induced myeloma cell apoptosis regardless of the sensitivity levels (*Figure 8B*, *Figure 8—source data 2*). To determine whether the expression of TRAIL receptors determines their relative sensitivities towards TRAIL, we determined the cell surface expression levels of DR4 and DR5. It turns out that both receptors are abundantly expressed in all three cell lines. For DR5, the highest expression was observed in IM-9 (32 x background), followed by RPMI-8226 (10 x background), and U266 (5 x background). For DR4, the highest expression was observed in RPMI8226 (26 x background), followed by U266 (18 x background) and IM-9 (5 x background) (*Figure 8C*). Thus, the expression level of TRAIL receptors may not be a major determining factor for the dramatic differences in sensitivity among these lines.

Next, we examined the abundance of cell surface HS by using a monoclonal anti-HS antibody (HS20), which preferably recognizes highly sulfated HS (*Gao et al., 2016*). Here, we found while RPMI-8226 and U266 cells both express abundant HS (*Figure 8D*), IM-9 cells express very small amounts of highly sulfated HS at the cell surface (1.5 x background). Because syndecan-1 has been shown to be the predominant HS proteoglycan expressed by most myeloma cells (*Sanderson and Yang, 2008*), we performed FACS analysis of cell surface syndecan-1. Interestingly, while syndecan-1 is abundantly expressed by both U266 and RPMI8226 cells, only a portion of IM-9 cells express syndecan-1 at a much lower level (*Figure 8E*). These observations suggest that reduced syndecan-1 expression could be the main contributor to the limited cell surface presentation of HS.

To examine whether the overall biosynthesis of HS differs among these three myeloma lines, we quantified the total amounts and disaccharide compositions of HS expressed by these cells. We found that the total amount of HS expressed by IM-9 cells is substantially lower, amounting to only 36% and 23% of total HS expressed by U266 and RPMI-8226 cells, respectively (*Table 3*). In sum, our result suggests that the greatly reduced syndecan-1 expression, combined with a reduced overall production of HS, contributes to the dramatic reduction of cell surface expression of HS in IM-9 cells, which might be a contributing factor to the resistance displayed by this cell line.

With regard to the difference in sensitivity between RPMI-8226 and U266 cells, cell surface HS expression level and sulfation level might also play a role based on two observations. First, the amount of highly sulfated HS expressed by RPMI8226 cells is 2.2-fold higher than U266 cells (*Table 3*). Second, cell surface HS staining suggests that the RPMI8226 cells display highly homogeneous expression of highly sulfated HS structures (*Figure 8D*, HS-20 staining gave a narrow peak), while the HS expression patten of U266 cells are more heterogeneous, including both high-expression and medium expression cells (*Figure 8D*, HS-20 staining gave a very broad peak).

## HS forms a complex with TRAIL and DR5 and regulates TRAIL-induced DR5 internalization

To have a more complete understanding of the role of HS in regulating TRAIL signaling, we investigated whether HS directly interacts with TRAIL receptor. When recombinant extracellular domain of DR5 was applied onto heparin Sepharose column, we found that no DR5 was retained on the heparin column (*Figure 9A*, left half, *Figure 9—source data 1*), suggesting there is no direct binding between DR5 and HS. However, when DR5 was premixed with TRAIL and then applied onto heparin column, we found DR5 and TRAIL both bound heparin column and were co-eluted in 500 mM and 1 M salt fractions (*Figure 9A*, right half, *Figure 9—source data 1*). This result strongly suggests that TRAIL-DR5 interaction and TRAIL-HS interaction are fully compatible with each other, which is consistent with the fact that the DR5 binding site and the HS-binding sites are spatially separated (*Figure 9B*). This finding raised the possibility that HS, a molecule that is commonly involved in the internalization of HS-binding proteins (*Christianson and Belting, 2014*; *Payne et al., 2007*), might regulate the internalization of DR5 after it is bound by TRAIL. On RPMI-8226 cells, we found that DR5 undergoes rapid internalization after stimulation with TRAIL (*Figure 9C*), while the internalization of DR4 is very

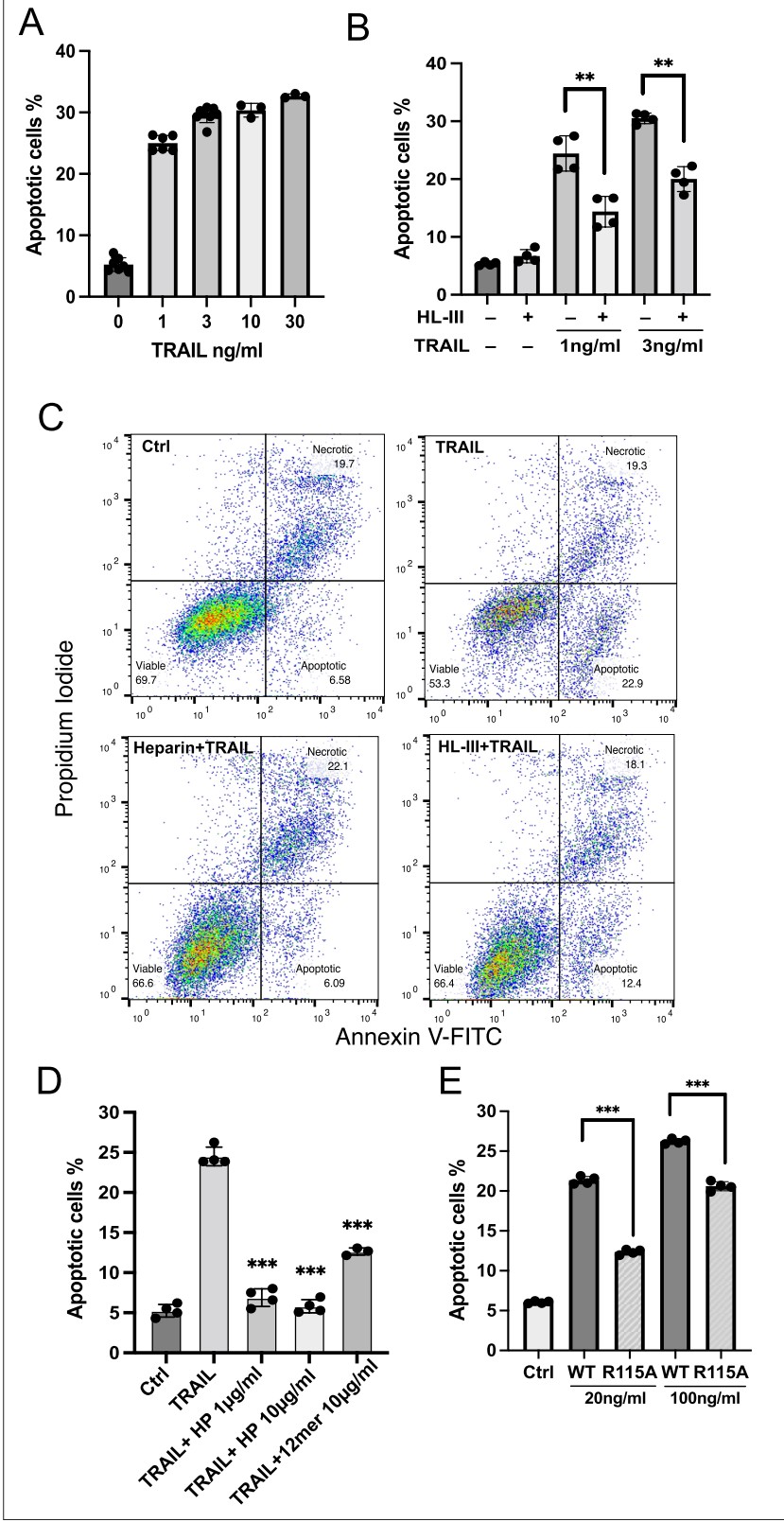

**Figure 7.** Cell surface heparan sulfate (HS) promotes TRAIL-induced myeloma cell apoptosis. (**A**) TNF-related apoptosis-inducing ligand (TRAIL)-induced apoptosis (at 1, 3, 10, 30 ng/ml) was tested using RPMI-8226 myeloma cells by Annexin V-FITC assay. Cells were analyzed after incubation with human TRAIL (hTRAIL) for 3 hr. (**B**) hTRAIL (1 or 3 ng/ml)- induced RPMI-8226 cell apoptosis with or without heparin lyase III (HL-III) treatment (5 mU/ml)

*Figure 7 continued on next page*

*Figure 7 continued*

were determined by Annexin V-FITC assay. (**C**) Representative scatter plots and gating method of Annexin V-FITC assay. Cells are treated with 1 ng/ml TRAIL in the presence of absence of HL-III (5 mU/ml) and heparin (1 µg/ml). (**D**) The effects of heparin and HS oligosaccharides 12mer towards TRAIL-induced RPMI8226 cell apoptosis were determined by Annexin V-FITC assay. (**E**) WT hTRAIL and R115A hTRAIL-induced RPMI8226 cell apoptosis were determined by Annexin V-FITC assay. Error bars represent S.D. ** represents p<0.01, *** represents p<0.0001 by Student's t-test. Data are representative of at least three separate assays.

The online version of this article includes the following source data for figure 7:

**Source data 1.** Excel file with raw data used to generate *Figure 7A*.

**Source data 2.** Excel file with raw data used to generate *Figure 7B*.

**Source data 3.** Excel file with raw data used to generate *Figure 7D*.

**Source data 4.** Excel file with raw data used to generate *Figure 7E*.

limited (*Figure 9D*). Focusing on TRAIL-induced DR5 internalization, we compared the internalization kinetics between intact cells and cells pre-treated with HL-III from 15 min to 1 hr. Interestingly, the removal of cell surface HS significantly reduced the rate of DR5 internalization (*Figure 9E*). By 60 mins, the internalization level of DR5 on HL-III treated cells was less than the internalization level of DR5 of cells without HL-III treatment at 15 min, which represents a reduction of internalization rate of at least fourfold after removal of cell-surface HS. This result suggests HS might regulate TRAIL signaling by altering TRAIL-induced DR5 internalization.

## Discussion

In this report, we identified TRAIL as a second member of the TNF superfamily that interacts with cell surface HS. Previously, it was reported that APRIL (A proliferation-inducing ligand) interacts with HS and the interaction plays an important role in APRIL localization and signaling (*Hendriks et al., 2005*; *Huard et al., 2008*; *Ingold et al., 2005*). Interestingly, many similarities were found between the TRAIL-HS and APRIL-HS interactions. First, the HS-binding site of TRAIL and APRIL are similar—both utilize three basic residues located at the N-terminal ends of the soluble forms (*Hendriks et al., 2005*). Due to the trimeric nature of TNF family proteins, three basic residues from three monomers can combine into a HS-binding site that comprises nine basic residues. Second, the binding of HS is fully compatible with the binding of TRAIL and APRIL to their respective receptors (*Hendriks et al., 2005*). At last, it was proposed that one mechanism by which HS promotes APRIL signaling is to promote its oligomerization at cell surface, which is also similar to our observation of HS-induced TRAIL oligomerization (*Kimberley et al., 2009*).

Our finding suggests that HS can regulate the biological function of TRAIL on several different levels. First, we have shown that HS can induce TRAIL to form higher-order oligomers. It is well-known that the clustering of TRAIL greatly promotes the apoptotic response through gathering large numbers of TRAIL receptors (*Montinaro and Walczak, 2023*; *von Karstedt et al., 2017*). Apparently, one way HS could promote TRAIL signaling is by clustering TRAIL on the tumor cell surfaces. As we have observed in MDA-MB-453 cells, which express low levels of DR5 (*Chen et al., 2012b*), heparin lyase treatment completely blocked TRAIL-induced apoptosis (*Figure 6B*). In contrast, in tumors cells that express high levels of TRAIL receptors, heparin lyase treatment only blocked 50–60% of TRAIL-induced apoptosis (*Figures 7B and 8B*). This difference suggests that the global contribution of HS-induced TRAIL-oligomerization to apoptosis likely depends on the expression level of TRAIL receptors. Due to the low density of TRAIL receptors on MDA-MB-453 cell surface, TRAIL-induced apoptosis has a greater dependence on HS to cluster TRAIL with its receptor.

A second way HS could regulate the function of TRAIL is to promote TRAIL-induced receptor internalization. While the exact role of TRAIL-induced internalization in apoptosis remains controversial, and the role could be very different in different tumor cells (*Kahraman et al., 2009*; *Artykov et al., 2021*; *Mazurek et al., 2012*; *Zhang et al., 2009*), our study in myeloma cells clearly shows that HS promotes rapid internalization of DR5 upon binding with TRAIL. The absence of HS at the cell surface greatly reduced the internalization rate for all time points (*Figure 9E*). While heparan sulfate proteoglycans (HSPGs) are known to play an active role in internalizing many HS-binding proteins (*Christianson and*

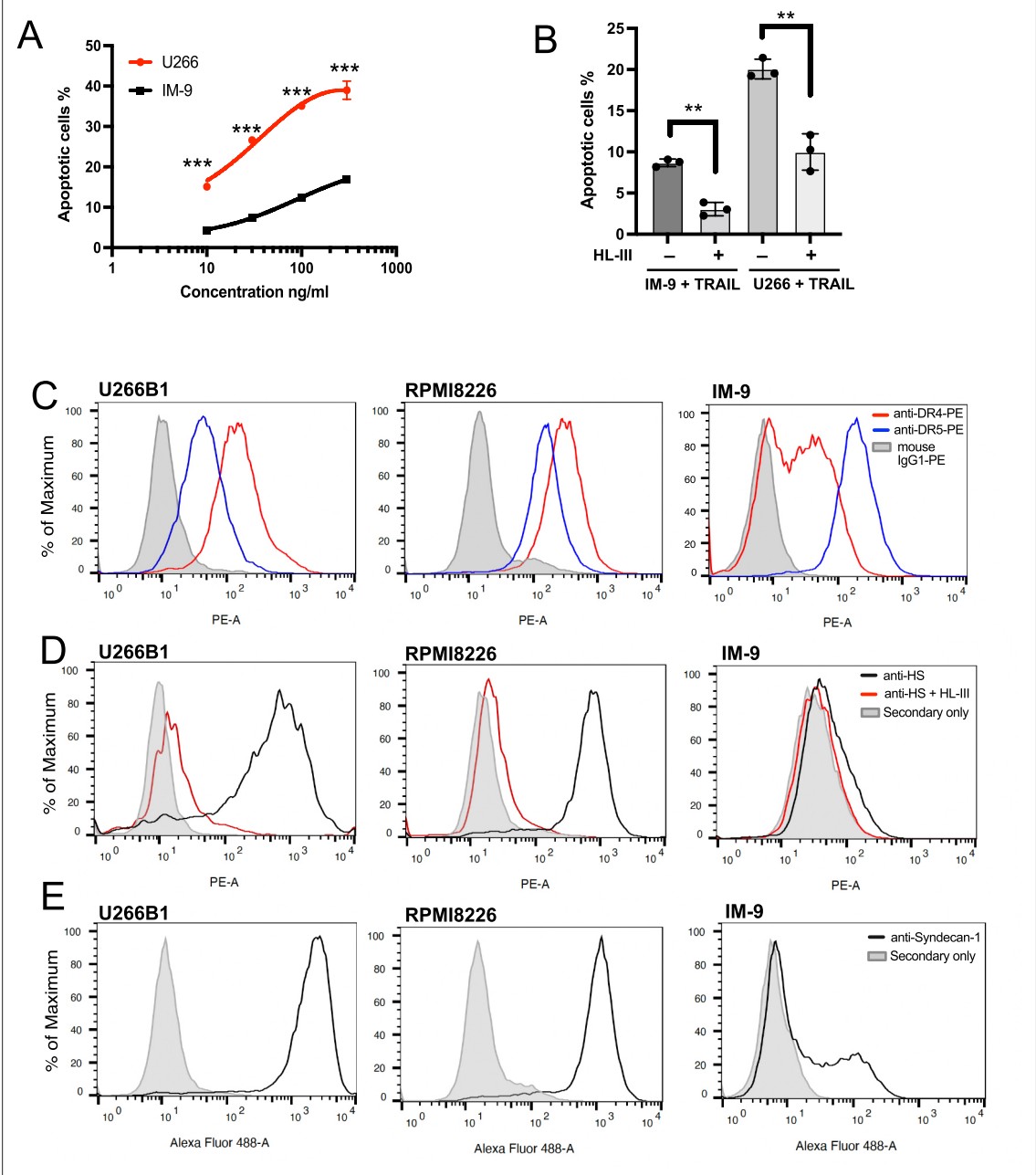

**Figure 8.** Cell surface heparan sulfate (HS) level contributes to TNF-related apoptosis-inducing ligand (TRAIL) sensitivity towards myeloma cells. (**A**) The sensitivity of U266 and IM-9 cells to TRAIL (10, 30, 100, 300 ng/ml) was determined by Annexin V-FITC assay. (**B**) TRAIL (100 ng/ml)-induced apoptosis of IM-9 and U266 cells, with or without heparin lyase III (HL-III) pretreatment, were determined by Annexin V-FITC assay. (**C**) Expression of TRAIL receptor DR4 and DR5 on U266B1, RPMI-8226, and IM-9 cells were determined with PE conjugated mAbs against DR4 and DR5 using FACS. The shaded histograms are from cells stained with mouse IgG1-PE conjugate. (**D**) Expressions of cell surface HS on untreated cells or cells pretreated with HL-III were determined by a human anti-HS mAb, followed by an anti-human-IgG Alexa-594 secondary antibody. The shaded histogram is from cells stained with secondary antibody only. (**E**) Expression of cell surface syndecan-1 was determined by a mouse anti-syndecan-1 mAb followed by an anti-mouse-IgG Alexa-488 secondary antibody. The shaded histogram is from cells stained with secondary antibody only. Error bars represent S.D. ** represents p<0.01 and *** represents p<0.0001 by Student's t-test. Data are representative of at least three separate assays.

The online version of this article includes the following source data for figure 8:

**Source data 1.** Excel file with raw data used to generate *Figure 8A*.

**Source data 2.** Excel file with raw data used to generate *Figure 8B*.

**Table 3.** Compositional analysis of heparan sulfate (HS) expressed by myeloma cell lines.

| Disaccharides | ng/10⁶ cells | | |
| --- | --- | --- | --- |
| | **IM-9** | **RPMI 8226** | **U266** |
| △UA2S-GlcNS6S | 1.53 | 4.16 | 1.34 |
| △UA-GlcNS6S | 0.26 | 5.12 | 0.78 |
| △UA2S-GlcNS | 1.38 | 2.43 | 3.30 |
| △UA-GlcNS | 2.66 | 14.33 | 9.74 |
| △UA2S-GlcNAc6S | 0.07 | 0.16 | 0.09 |
| △UA-GlcNAc6S | 0.26 | 7.57 | 0.63 |
| △UA2S-GlcNAc | 0.59 | 0.76 | 0.86 |
| △UA-GlcNAc | 19.38 | 77.95 | 54.70 |
| Total amount of HS | 26.13 | 112.48 | 71.44 |
| Disaccharides with ≥2 sulfations | 3.24 | 11.87 | 5.51 |

*Belting, 2014*; *Payne et al., 2007*), the unique finding here is that, although DR5 is not a HS-binding protein itself, its internalization is HS-dependent. This dependence can be explained by our finding that DR5 becomes associated with HSPGs through TRAIL by forming a ternary complex (*Figure 9A*). Interestingly, our lab previously reported another similar ternary complex involving RANKL, which also belongs to TNF superfamily (*Li et al., 2016*). But in that ternary complex, RANKL (not a HS-binding protein) becomes associated with HS through its HS-binding decoy receptor OPG.

In this study, we utilized several biophysical methods to gain structural insights of HS-TRAIL interaction. By SEC-MALS and SAXS analysis of TRAIL-oligosaccharide complexes, we confirmed that HS 18mer (*Figures 2B and 4A*) can indeed induce TRAIL to form a stable homohexamer. While the hexamer model generated from the SAXS data indicates the hexamer may exist as a dimer of trimers (*Figure 4B*), due to the round shape of the TRAIL homotrimer, more precise determination of the relative orientation of the two trimers was not possible. We have also attempted to solve the co-crystal structure of the complex between mTRAIL and HS 12mer, but the crystal we obtained unfortunately did not contain HS 12mer. However, from this crystal structure we discovered that mTRAIL exists in a strand-swapped homotrimer. Close examination of several crystal structures of human TRAIL found that a significant portion (11–14 aa) of a long loop connecting strands A and A" is disordered in a number of the structures (*Cha et al., 1999*; *Cha et al., 2000*; *Hymowitz et al., 1999*; *Hymowitz et al., 2000*), which raises the possibility that in some conditions, hTRAIL might also exist in strand-swapped trimers. Whether a strand swap is a regulatory component of TRAIL's function remains to be seen and warrants further investigation.

A main obstacle facing TRAIL-based anti-tumor therapy is tumor resistance to TRAIL. Based on current understanding, the main contributing factors for resistance are first, the overall expression of TRAIL receptors; and second, the expression levels of essential components of intracellular apoptotic proteins such as caspases (*Montinaro and Walczak, 2023*; *von Karstedt et al., 2017*). Our finding strongly suggests that a third contributor to TRAIL resistance could be the expression level and composition of HS on tumor cell surfaces. The fact that removal of cell surface HS greatly diminishes TRAIL-induced apoptosis in both breast cancer and myeloma lines already suggests that altering cell surface presentation of HS could have a substantial effect on tumor sensitivity to TRAIL. To investigate the potential correlation of HS expression level and TRAIL-sensitivity, we examined three myeloma cell lines with different sensitivities to TRAIL, with RPMI-8226 as the most sensitive line (*Figure 7A*), U266 as the moderately sensitive line and IM-9 as the most resistant line (*Figure 8A*). Surprisingly, our analysis found that RPMI8226 cells express the highest amounts of HS and syndecan-1, while IM-9 express the least amount of HS and syndecan-1 (*Figure 8D&E*, *Table 3*). In other words, in this set of myeloma cells we found a strong correlation between HS/HSPG levels and sensitivities to TRAIL. Our data suggests that the expression level of HS/HSPG can be one important compounding factor that contributes to the sensitivity of tumor cells to TRAIL.

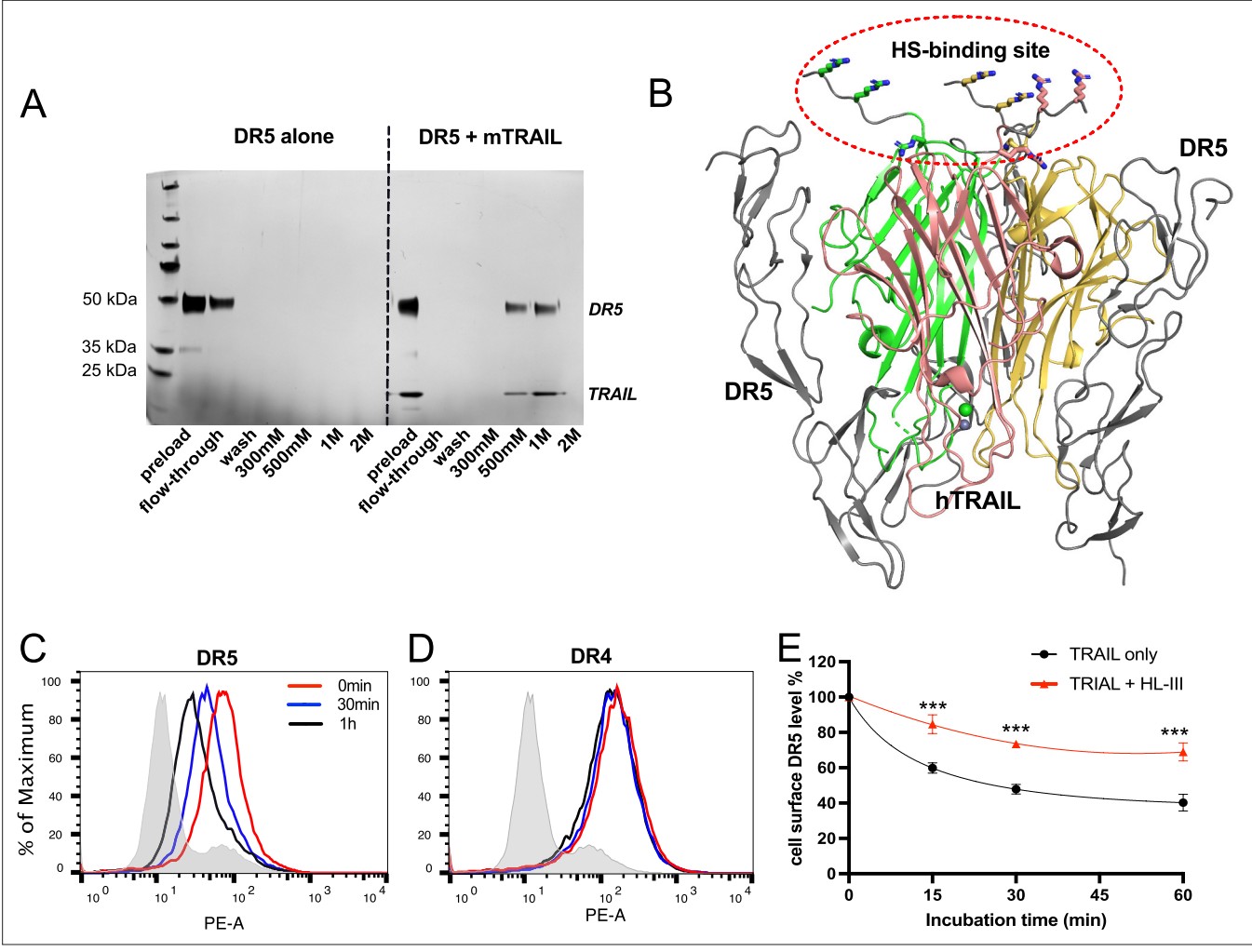

**Figure 9.** Heparan sulfate (HS) forms a complex with TNF-related apoptosis-inducing ligand (TRAIL) and death receptor 5 (DR5) and regulates TRAIL-induced DR5 internalization. (**A**) While DR5 does not bind heparin by itself (left half of the gel), DR5-TRAIL complex can bind heparin (right half of the gel), indicating DR5-TRAIL-heparin can form a ternary complex through TRAIL. Representative of three experiments with identical results (**B**) DR5 and HS bind to different surfaces on TRAIL. Crystal structure of hTRAIL-DR5 complex (IDU3). Human TRAIL (hTRAIL) is shown in the cartoon and the three monomers are displayed in green, salmon and gold, respectively. The three DR5 molecules are shown in gray cartoon. Because residues 114–119 of hTRAIL are disordered in this structure, these residues (114VRERGP119, backbone shown in gray random coils) were manually modeled onto the last visible N-terminal residue (Q120) of the hTRAIL. Sidechains responsible for HS binding (from R115, R117, and R121) are shown in sticks. (**C–E**) TRAIL-dependent internalization of DR4 and DR5 was determined by a FACS-based assay. Cell surface levels of TRAIL receptor DR5(C) and DR4 (**D**) were determined before TRAIL stimulation, and 30 min and 1 hr after TRAIL stimulation. The shaded histograms are from cells stained mouse IgG1-PE conjugate. (**E**) Plot of time-dependent internalization of cell surface DR5, with or without HL-III treatment. n=3. *** represents p<0.0001. Data is representative of three experiments with similar results.

The online version of this article includes the following source data for figure 9:

**Source data 1.** Original silver stain gel picture for *Figure 9A*.

**Source data 2.** Original silver stain gel picture for *Figure 9A* with relevant lanes labeled.

**Source data 3.** Excel file with raw data used to generate *Figure 9E*.

The potent inhibitory effect of exogenous heparin on TRAIL-induced apoptosis is quite intriguing and may have important implications in clinical setting when TRAIL-based anti-tumor therapy is attempted (*Figures 6B and 7D*). Due to greatly increased risk of venous thrombosis, it is common for cancer patients to receive heparin as an thromboprophylaxis (*Lee and Peterson, 2013*; *Pernod et al., 2020*). In fact, daily administration of LMWH for 6 months has been a standard treatment for cancer-associated thrombosis since early 2000. This practice raised an important question: when patients receive heparin and TRAIL-based therapy at the same time, will circulating heparin diminishes the

effect of TRAIL? Our data suggest that the use of heparin may complicate evaluating the efficacy of TRAIL therapy in clinical trials.

In conclusion, our study has provided strong evidence that HS plays an essential roles in TRAIL-induced tumor cell apoptosis. These new mechanistic insights will promote a more comprehensive understanding of TRAIL biology and may one day lead to novel TRAIL-based anti-tumor therapy.

# Materials and methods

**Key resources table**

| Reagent type (species) or resource | Designation | Source or reference | Identifiers | Additional information |
|---|---|---|---|---|
| Strain, strain background (*Escherichia coli*) | Origami-B (DE3) | Millipore Sigma | Cat#: 70837 | |
| Cell line (*Homo-sapiens*) | MDA-MB-453 | ATCC | Cat#: HTB-131 | Identity authenticated by SRT profiling, negative for mycoplasma |
| Cell line (*Homo-sapiens*) | RPMI-8226 | ATCC | Cat#: CCL-155 | Identity authenticated by SRT profiling, negative for mycoplasma |
| Cell line (*Homo-sapiens*) | U266B1 | ATCC | Cat#: TIB-196 | Identity authenticated by SRT profiling, negative for mycoplasma |
| Cell line (*Homo-sapiens*) | IM-9 | ATCC | Cat#: CCL-159 | Identity authenticated by SRT profiling, negative for mycoplasma |
| Antibody | anti-mouse TRAIL (goat polyclonal) | R&D systems | Cat#: AF1121 | FC: 1 µg/ml |
| Antibody | anti-human TRAIL (goat polyclonal) | R&D systems | Cat#: AF375 | FC: 1 µg/ml |
| Antibody | Anti-human DR4 (mouse monoclonal) | Biolegend | Cat#: B376455 | FC: 1 µg/ml |
| Antibody | Anti-human DR5 (mouse monoclonal) | Biolegend | Cat#: B347680 | FC: 1 µg/ml |
| Antibody | Anti-HS human monoclonal (HS-20) | PMID:27185050 | Gift from Dr. Ho (NCI) | FC: 2 µg/ml |
| Antibody | Anti-human syndecan-1 (mouse monoclonal) | Biolegend | Cat#: B280312 | FC: 10 µg/ml |
| Recombinant DNA reagent | pET21b (plasmid) | Millipore Sigma | Cat#: 69741 | |
| Recombinant protein | Murine TRAIL (*E. coli*) | Produced in the lab | | |
| Recombinant protein | Human TRAIL (*E. coli*) | Produced in the lab | | |
| Recombinant protein | Human TRAIL (mammalian) | Biolegend | B294007 | |
| Recombinant protein | Mouse DR5-Fc fusion | R&D systems | 721-DR | |
| Commercial assay or kit | TUNEL staining kit | Thermofisher | Cat#: C10619 | |
| Commercial assay or kit | Annexin-FITC staining kit | R&D Systems | Cat#: 4830–250 K | |
| Software, algorithm | ImageJ (v1.50i) | PMID:22930834 | RRID:SCR_003070 | |
| Software, algorithm | Graphpad Prism 7 | GraphPad Software | RRID:SCR_002798 | |

## Expression and purification of the extracellular domain of mouse TRAIL (aa118-291) and human TRAIL (aa115-281) in *E. coli*

Recombinant mouse or human TRAIL was generated in *Escherichia coli*. The coding sequence of mouse TRAIL (aa118-291) or human TRAIL (aa115-281) was amplified from its cDNA and was cloned into pET21b (Novagen) using NdeI and XhoI sites. Expression was carried out at 18 °C in Origami-B cells (Novagen) carrying the pGro7 (Takara) plasmid expressing chaperonin proteins GroEL and GroES following an established protocol (*Zhang et al., 2021*). Purification was carried out using HiTrap SP cation exchange chromatography (with Buffer A: 25 mM MES, pH 6.5, 50 mM NaCl; and Buffer B: 25 mM MES, pH 6.5, 1 M NaCl), followed by gel permeation chromatography with HiLoad 16/60

Superdex 200 (GE healthcare) in 25 mM HEPES, pH 7.1, 150 mM NaCl. After purification, TRAIL was >99% pure, as judged by silver staining.

## Heparin–sepharose chromatography

To characterize the binding of WT TRAIL and TRAIL mutants to heparin, 100 µg of purified WT or mutant TRAIL was applied to a 1 ml HiTrap heparin–Sepharose column (Cytiva Lifesciences) and eluted with a salt gradient from 150 mM to 1 M NaCl at pH 7.1 in 25 mM HEPES buffer. The conductivity measurements at the peak of the elution were converted to the concentration of NaCl based on a standard curve.

## Site-directed mutagenesis

TRAIL mutants were prepared using a previously published method (*Zheng et al., 2004*). Mutations were confirmed by Sanger sequencing, and recombinant protein was expressed as described for WT TRAIL. Purification was carried out using HiTrap SP cation exchange column at pH 7.1 (HEPES buffer), followed by gel permeation chromatography as described for WT TRAIL.

## Surface plasmon resonance

SPR was performed on an OpenSPR instrument (Nicoya). Biotinylation of chemoenzymatically synthesized HS 12merNS2S6S was prepared as previously described (*Arnold et al., 2020*). Biotinylated 12merNS2S6S was immobilized to a streptavidin sensor chip (Nicoya) based on the manufacturer's protocol. In brief, 150 µl of solution of the biotin-12merNS2S6S (18 µg/ml) in HBS-running buffer (25 mM HEPES, pH 7.1, 150 mM NaCl, 0.05% Tween-20) was injected to channel 2 of the flow cell of the sensor chip at a flow rate of 20 µl/min. The successful immobilization of biotin-12merNS2S6S was confirmed by the observation of a 100–200 resonance unit increase in the sensor chip. The flow cell channel 1 was used as the background control, which was not immobilized with biotin-12merNS2S6S. Different dilutions of hTRAIL (concentrations from 68 to 1088 nM) in HBS-running buffer were injected at a flow rate of 20 µl/min. At the end of the sample injection, the same buffer was flowed over the sensor surface to facilitate dissociation. After a 5 min dissociation time, the sensor surface was regenerated by injecting with 150 µl regeneration buffer (25 mM HEPES, pH 7.1, 2 M NaCl) at a flow rate of 150 µl/min to get a fully regenerated surface. The sensorgrams were fit with 1:1 Langmuir binding model from TraceDrawer 1.9.2.

## Analytical size-exclusion chromatography (SEC)

For analysis of mTRAIL and HS oligosaccharide complexes, purified mTRAIL (100 µg) was incubated with HS oligosaccharides (molar ratio 1:1) in 25 mM HEPES, 150 mM NaCl, pH 7.1, at room temperature for 1 hr. For analysis of mTRAIL and low molecular weight/full-length heparin complex, purified mTRAIL (100 ug) was incubated with heparin (molar ratio 1:1) in 25 mM HEPES, 150 mM NaCl, pH 7.1, at room temperature for 1 hr. All complexes were resolved on a Superdex 200 Increase filtration column (Cytiva Lifesciences) using 25 mM HEPES, 150 mM NaCl, pH 7.1, at 4 °C. Presence of a para-nitrophenyl group in the reducing end of the oligosaccharides allows excess oligosaccharides to be visible in the A280 elution profile.

## SEC-multiangle light scattering (MALS)

SEC-MALS analysis was performed using a DAWN MALS detector (Wyatt Technology) connected to an AKTA FPLC system (GE Healthsciences). mTRAIL, mTRAIL/12mer complex and mTRAIL/18mer complex were prepared as describe above and concentrated to ~4 mg/ml, and 100 µl was resolved on Superdex 200 Increase SEC column using 25 mM HEPES, 150 mM NaCl, pH 7.1. The MALS data was analyzed using ASTRA software (ver. 7.3.2.17).

## Crystallization of mTRAIL

TRAIL was crystallized using the sitting drop vapor diffusion technique by mixing 400 nl of protein solution consisting of 4.4 mg/ml TRAIL, 1 mM HS 12merNS2S6S, 25 mM HEPES pH 7.1, and 150 mM NaCl with 250 nl of the reservoir consisting of 85 mM HEPES pH 7.5, 8.5% PEG 8000 (w/v), and 8.5% ethylene glycol (v/v). Crystals were harvested by adding 1 ul of cryo solution, consisting of 90% reservoir and an additional 10% ethylene glycol, directly to the crystal drop prior to mounting the crystal

and flash freezing in liquid nitrogen. Data were collected on the Southeast Regional Collaborative Access Team (SER-CAT) 22-ID beamline at the Advanced Photon Source, Argonne National Laboratory (*Table 2*). Data were integrated and scaled using HKL2000 (*Otwinowski and Minor, 1997*). The structure was solved by performing molecular replacement using PDB coordinates 1DU3 *Cha et al., 2000* followed by iterative cycles of refinement in Phenix and manual model building in Coot (*Adams et al., 2010*; *Emsley and Cowtan, 2004*; *Emsley et al., 2010*; *Zwart et al., 2008*). Model statistics and quality were evaluated using MolProbity (*Chen et al., 2010*) and are presented in (*Table 2*).

## Small-angle X-ray scattering

Scattering data were collected at the beamline 12.3.1 at the Lawrence Berkeley National Laboratory using a SEC-SAXS mail-in service (*Classen et al., 2013*). mTRAIL-12mer and mTRAIL-18mer complexes were prepared by mixing 1 mg mTRAIL with 160 µg 12mer or 240 µg 18mer (molar ratio 1:1 for both) for 1 h in HEPES buffer (25 mM HEPES, pH 7.1, 0.15 M NaCl, pH 7.1) at room temperature. The complexes, and free mTRAIL, were concentrated to 6 mg/ml for data collection. The proteins were resolved on an SEC column (Protein KW-802.5, Shodex) on an Agilent 1260 series HPLC. SAXS data were collected from in-line eluent as the samples come off the column, and 3 s exposures were collected for each frame over the course of 33 min (~660 frames for the entire run). The scattering data of the frames corresponding to the protein peak (10 frames) were averaged and used for data analysis. Determination of Guinier plot, P(r) function plot, and MW estimation was performed using RAW (version 2.14) (*Hopkins et al., 2017*; *Putnam et al., 2007*). The *ab initio* model of the mTRAIL-18mer complex was generated by GASBOR (*Svergun et al., 2001*), based on P2 symmetry. The whole set of experiments were performed twice using two different mTRAIL preparations with similar results.

## Fluorescence-activated cell sorting

4T1 cells and MDA-MB-453 cells were incubated with 100 or 300 ng/mL mTRAIL or hTRAIL, respectively, in PBS containing 0.1% BSA for 1 hr at 4 °C. Bound TRAIL was stained with goat anti-mTRAIL (1 ug/ml, AF1121, R&D systems) or goat anti-hTRAIL (1 ug/ml, AF375, R&D systems) for 1 hr at 4 °C, followed by anti-goat IgG–Alexa647 (1:1000, Thermo Fisher Scientific) for 30 min, and washed and fixed in 2% PFA for flow cytometry analysis. The cell surface expression levels of DR4 and DR5 on RPMI8226, U266B1, and IM-9 cells were evaluated directly with PE-conjugated anti-human DR4 antibody (B376455, Biolegend) and PE-conjugated anti-human DR5 antibody (B347680, Biolegend), both at 1 µg/ml. For control, cells were stained with PE-conjugated Rat IgG1. The cell surface expression level of HS on RPMI-8226, U266B1, and IM-9 cells were evaluated by incubating cells with 2 µg/ml HS20 antibody (gift from Dr. Michell Ho, NCI) for 1 hr at 4 °C, followed by anti-human IgG–Alexa594 (1:1000, Thermo Fisher Scientific) for 30 min and analyzed by flow cytometry. For control, cells were stained with anti-human IgG–Alexa594 only. The cell surface expression level of syndecan-1 was evaluated by incubating cells with anti-syndecan-1 antibody (10 µg/ml, B280312, Biolegend) for 1 hr at 4 °C, followed by anti-mouse IgG–Alexa488 (1:1000, Thermo Fisher Scientific) for 30 min and analyzed by flow cytometry. For control, cells were stained with anti-mouse IgG–Alexa488 only. In some experiments, cells were pretreated with recombinant HL-III (5 milliunits/ml, produced in our lab) for 15 min at room temperature prior to binding experiments.

## Annexin V-FITC assay

Apoptosis of tumor cells was determined with an Annexin-FITC staining kit (R&D Systems) as per the manufacturer's instructions and analyzed by flow cytometry. Briefly, breast cancer cells and myeloma cells were treated with recombinant hTRAIL (B294007, Biolegend, 1–300 ng/ml for different cells) for 3 hr in the presence or absence of either HL-III (5 milliunits/ml), heparin (1–20 µg/ml) or HS 12mer (10 µg/ml). The treated cells were harvested and incubated with a reaction mixture containing Annexin-FITC and Propidium iodide in the dark for 15–30 min for labeling. Annexin-FITC binds phosphatidylserine that are exposed in apoptotic cells. Propidium iodide (PI) is a membrane-impermeant DNA dye and only stains cells that have lost membrane integrity (necrotic cells) but not early apoptotic cells. Fluorescence of PI often increase at least 30-fold after binding to DNA. In our hands, after HL-III or heparin treatments the PI signal often decreased ~twofold on viable cells (*Figures 6A and 7C*). Because PI is positively charged, it is likely that cell surface HS binds PI and such interaction might contribute to the background staining of PI on live cells. We also observed viable cells that were

treated with TRAIL alone had a 1.5–2 fold increase in PI signal. The reason for this increase is unknown. In all experiments, early apoptotic cells (labeled apoptotic cells for simplicity) were identified as PI low (in the PE channel), Annexin-FITC high cells; while necrotic cells were identified as PI high, Annexin-FITC high cells.

## TUNEL assay

MAD-MB-453 cells were treated with recombinant hTRAIL (30 ng/ml, B294007, Biolegend) for 6 hr, in the presence of absence of HL-III (5 milliunits/ml) or heparin (1 µg/ml). TUNEL staining was performed using an apoptosis terminal deoxynucleotidyl transferase (TdT) DNA fragment detection kit (ThermoFisher), according to the manufacturer's instructions. Briefy, cells were fixed with 4% paraformaldehyde in PBS and permeabilized with 0.25% Triton X-100. Cells were then incubated with TdT reaction mixture for 60 min at 37 °C, followed by incubation with Click-iT Plus TUNEL reaction cocktail (with Alexa647 dye) for 30 min at 37 °C. Slides were mounted with Prolong mounting medium with DAPI and images were taken with a Nikon Ci-S fluorescence microscope.

## Disaccharide analysis of myeloma cell HS

### Sample preparation

RPMI-8226, U266B1 and IM-9 cells were cultured to subconfluence in RPMI-1640 medium supplemented with 15% FBS, and $8 \times 10^6$ cells were harvested to purify cellular HS. Cell pellets were resuspended in 750 µL of water and digested with 150 µL pronase E (20 mg/mL, Sigma-Aldrich) at 55 °C for 24 hr. After proteolyzed, the solution was boiled at 100 °C for 10 min, and centrifuged at 14,000 rpm for 10 min. Before loading to DEAE column, 2 µL $^{13}$C-labeled N-sulfated K5 polysaccharide (45 ng/µl) was added to the supernatant. DEAE column buffer A contained 20 mM Tris, pH 7.5, and 50 mM NaCl, and buffer B contained 20 mM Tris, pH 7.5 and 1 M NaCl. After loading the sample into the DEAE column, the column was washed with 1.5 mL buffer A, followed by 1.5 mL buffer B to elute the HS. The eluted HS was desalted using an YM-3-kDa spin device using deionized water, and the desalted HS was dried for heparin lyases digestion. Samples were digested in 100 µL heparin lyases digestion buffer (100 mM sodium acetate, 2 mM calcium acetate buffer, and 0.1 g/L BSA, pH 7.0) containing heparin lyase I (60 µg/ml), II (340 µg/ml), and III (500 µg/ml). The digestion solution was incubated at 37 °C for 12 hr, after which it was boiled at 100 °C for 10 min. Before recovering the digests from the digest solution, a known amount of $^{13}$C-labeled disaccharide calibrants ($\triangle$[$^{13}$C]UA-GlcNAc, $\triangle$[$^{13}$C]UA2S-GlcNAc, $\triangle$[$^{13}$C]UA-GlcNAc6S, $\triangle$[$^{13}$C]UA2S-GlcNAc6S, $\triangle$[$^{13}$C]UA-GlcNS, $\triangle$[$^{13}$C]UA2S-GlcNS, $\triangle$[$^{13}$C]UA-GlcNS6S, and$\triangle$[$^{13}$C]UA2S-GlcNS6S) were added to the digestion solution. The HS disaccharides were recovered by centrifugation, and supernatant were freeze-dried before the AMAC derivatization.

### Chemical derivatization of HS disaccharides

The 2-Aminoacridone (AMAC) derivatization of lyophilized samples was performed by adding 10 µL of 0.1 M AMAC solution in DMSO/glacial acetic acid (17:3, v/v) and incubating at room temperature for 15 min. Then 10 µL of 1 M aqueous sodium cyanoborohydride (freshly prepared) was added to this solution. The reaction mixture was incubated at 45 °C for 2 hr. After incubation, the reaction solution was centrifuged to obtain the supernatant that was subjected to the LC-MS/MS analysis.

### LC-MS/MS analysis

The analysis of AMAC-labeled disaccharides was performed on a Vanquish Flex UHPLC System (Thermo Fisher Scientific) coupled with TSQ Fortis triple-quadrupole mass spectrometry as the detector. The C18 column (Agilent InfinityLab Poroshell 120 EC-C18 2.7 µm, 4.6 × 50 mm) was used to separate the AMAC-labeled disaccharides. Buffer A was 50 mM ammonium acetate in water and buffer B is methanol. The elution gradient was from 5–45% buffer B in 10 min, followed by 100% buffer B in 4 min, at a flow rate of 0.3 ml/min. Online triple-quadrupole mass spectrometry operating in the multiple-reaction-monitoring (MRM) mode was used as the detector. The ESI-MS analysis was operated in the negative-ion mode using the following parameters: Neg ion spray voltage at 4.0 kV, sheath gas at 45 Arb, aux gas 15 arb, ion transfer tube temp at 320 °C, and vaporizer temp at 350 °C. TraceFinder software was applied for data processing.

## Binding of DR5-TRAIL complex to heparin-sepharose

Recombinant mouse DR5 (aa53-177)-Fc fusion protein (721-DR, R&D) alone (10 μg), or DR5–mTRAIL complex (10 μg each pre-incubated for 1 hr at room temperature), were loaded onto heparin-Sepharose (Cytiva) gravity column (200 μl bed volume). Column was first washed with 2 ml buffer A (25 mM HEPES, pH7.1, 150 mM NaCl), followed by four elution steps (800 μl each) using buffers containing 300 mM, 500 mM, 1 M, and 2 M NaCl, respectively. 30 μl of eluents from each step were resolved on a 4–20% SDS-PAGE gel and the gel was visualized by silver staining.

## DR5 and DR4 internalization assay

RPMI-8226 cells were treated with 50 ng/ml recombinant hTRAIL at 37 C for 15, 30, or 60 min, Selected samples were pretreated with HL-III for 15 min prior to the addition of hTRAIL. Cell surface expression DR5 and DR4 were determined by flow cytometry as described above.

## Statistical analysis

All data are expressed as means ± SDs. Statistical significance was assessed using two-tailed Student's t-tests or analysis of variance (ANOVA) using GraphPad Prism software (GraphPad Sofware Inc). p--value <0.05 was considered significant.

## Acknowledgements

Use of the Advanced Photon Source was supported by the US Department of Energy, Office of Science, Office of Basic Energy Sciences, under Contract No. W-31–109-Eng-38. SAXS work was conducted at the Advanced Light Source (ALS), a national user facility operated by Lawrence Berkeley National Laboratory on behalf of the Department of Energy, Office of Basic Energy Sciences, through the Integrated Diffraction Analysis Technologies (IDAT) program, supported by DOE Office of Biological and Environmental Research. Additional support comes from the National Institute of Health project ALS-ENABLE (P30 GM124169) and a High-End Instrumentation Grant S10OD018483. The flow cytometry data in this study was acquired at the Optical Imaging and Analysis Facility, School of Dental Medicine, State University of New York at Buffalo. We also thank Dr. Mitchell Ho (NCI) for generously gifting us anti-HS mAb HS-20. The authors thank Andrea Kaminski and Lalith Perera for critical reading of the manuscript and Juno Krahn for structural discussions. This work was supported by R01AR070179 (YL, HH, CYO, DX), R01DE031273 (YL, HH, CYO, DX), GM142304 (ZW), R01HL094463 (JL), and in part by the Intramural Research Program of the National Institute of Environmental Health Sciences, NIH. ZIC ES102645 (LCP).

## Additional information

### Competing interests

Zhangjie Wang: is also an employee at Glycan Therapeutics. Yongmei Xu, Jian Liu: is a founder of Glycan Therapeutics and has stock ownership of the company. The other authors declare that no competing interests exist.

### Funding

| Funder | Grant reference number | Author |
|--------|------------------------|--------|
| National Institute of Arthritis and Musculoskeletal and Skin Diseases | R01AR070179 | Yin Luo Huanmeng Hao Chih Yean Ong Ding Xu |
| National Institute of Dental and Craniofacial Research | R01DE031273 | Yin Luo Huanmeng Hao Chih Yean Ong Ding Xu |
| National Institute of General Medical Sciences | GM142304 | Zhangjie Wang |

| Funder | Grant reference number | Author |
|---|---|---|
| National Heart, Lung, and Blood Institute | R01HL094463 | Jian Liu |
| National Institute of Environmental Health Sciences | ZIC ES102645 | Lars C Pedersen |

The funders had no role in study design, data collection and interpretation, or the decision to submit the work for publication.

### Author contributions

Yin Luo, Data curation, Formal analysis, Investigation, Visualization, Methodology, Writing - original draft; Huanmeng Hao, Zhangjie Wang, Data curation, Formal analysis, Investigation, Methodology; Chih Yean Ong, Data curation, Formal analysis, Investigation; Robert Dutcher, Data curation, Formal analysis, Methodology; Yongmei Xu, Resources, Methodology; Jian Liu, Resources, Supervision, Funding acquisition, Validation, Writing – review and editing; Lars C Pedersen, Conceptualization, Resources, Data curation, Formal analysis, Funding acquisition, Validation, Investigation, Visualization, Methodology, Writing – review and editing; Ding Xu, Conceptualization, Resources, Data curation, Formal analysis, Supervision, Funding acquisition, Validation, Visualization, Methodology, Writing - original draft, Project administration, Writing – review and editing

### Author ORCIDs

Robert Dutcher http://orcid.org/0000-0002-5144-2111
Ding Xu http://orcid.org/0000-0001-9380-2712

Reviewer #1 (Public Review): https://doi.org/10.7554/eLife.90192.3.sa1
Reviewer #2 (Public Review): https://doi.org/10.7554/eLife.90192.3.sa2
Author Response https://doi.org/10.7554/eLife.90192.3.sa3

## Additional files

### Supplementary files

• MDAR checklist

### Data availability

Figure source data contain the numerical data used to generate the figures.

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
