## [Editor Report · eLife assessment]

This **fundamental** study advances our understanding of TRAIL-induced apoptosis by defining how Heparan triggers this pathway at the molecular level. The evidence supporting the conclusions is **compelling**, with rigorous binding assays, structural methods, and cellular studies. The work will be of broad interest to cell biologists and biochemists.

---

## [Referee Report · Reviewer #1 (Public Review)]

Summary: TRAIL (Tumor necrosis factor (TNF)-related apoptosis-inducing ligand) is a potent inducer of apoptosis in tumor cells. Initially, this finding raised high expectations on the possibility to induce tumor-specific apoptosis by activation of TRAIL-receptors DR4 and DR5. However, attempted TRAIL-based anti-tumor therapies failed so far, and several tumor types were found to resist TRAIL-induced apoptosis. Yin Luo and colleagues provide an explanation for these observations with the potential to provide a new important biomarker for future TRAIL-based anti-tumor therapies and to reduce resistance. The authors reveal that sensitivity towards TRAIL correlates inversely with heparan sulfate (HS) expression levels at the surface of tumor cells, suggesting that HS functions as a tumor suppressor. These observations are explained by two two mechanisms: First, HS induces the assembly of higher-order oligomers from soluble TRAIL trimers, and second, TRAIL and HS form a ternary complex with DR5 to promote its cellular internalization. Therefore, this timely and important work provides a better mechanistic understanding of TRAIL-induced apoptosis and TRAIL resistance of some tumor types, with the potential to improve therapy.

Strengths: The major novel finding of this study is that extracellular heparan sulfate (HS) acts as a positive regulator of TRAIL-induced tumor cell apoptosis, and that HS expression of different tumor cell lines correlates with their capacity to induce cell death. The authors first show by affinity chromatography and SPR that murine and human TRAIL bind strongly to heparin (heparin is a highly sulfated, and thus strongly negatively charged form of HS that is derived from connective tissue type mast cells), and identify three basic amino acids on the TRAIL N-terminus that are required for the interaction. Size exclusion chromatography (SEC) and multiangle light scattering (MALS) revealed that TRAIL exists as a trimer that requires a minimum heparin length of 8 sugar residues for binding, and small angle X-ray scattering (SAXS) showed that TRAIL interaction with longer oligosaccharides induced higher order multimerization of TRAIL. Consistent with these biochemical and biophysical analyses, HS on tumor cells contributes to TRAIL-binding to their cell surface and subsequent apoptosis. The study also describes domain swapping observed by TRAIL trimer crystallization, and demonstrates different degrees of HS core protein and DR receptor expression in different tumor cell types. These findings are well supported and together with the advanced and established methodology used by the authors are the strengths of this paper. The paper will be of great interest to medical biologists studying TRAIL-resistance of tumors, to biologists interested in DR4 and DR5 receptor function and the effects of receptor internalization, and to glycobiologists aiming to understand the multiple important roles that HS plays in development and disease. The authors also raise the important point (and support it well) that routine heparin treatment of cancer patients potentially interferes with TRAIL-based therapies, providing one possible reason for their failure.

Weaknesses: Despite the importance and the clear strengths of the paper, some of its aspects could have been developed further. First, the authors findings that HS at the tumor surface promotes TRAIL binding, and that HS promotes TRAIL-induced breast cancer and myeloma cell apoptosis, are based on pre-treatment of cells with heparinase to remove surface HS prior to TRAIL-treatment, or on the addition of soluble heparin to compete with cell-surface HS for TRAIL binding. A more direct way to establish such new HS function could have been the genetic manipulation of cancer cells to overexpress HS or to express less or undersulfated HS. Changed susceptibility of these cells to TRAIL-induced apoptosis would have greatly underlined the physiological significance of the authors findings. Second, the mechanistics of TRAIL-induced, HS-modulated tumor cell apoptosis could have been more clearly defined. For example, the authors demonstrate convincingly that cell surface HS is essential for TRAIL-induced apoptosis in MDA-MB-453 breast cancer cells, and show that a tumor cell line (IM-9 cells) that expresses HS and the core protein to which HS is attached to only limited degrees is the most resistant to TRAIL-induced apoptosis. However, Indeed, the authors later also report that cell surface HS promotes TRAIL-induced myeloma cell apoptosis regardless of the sensitivity levels, and that other factors - the degree of TRAIL multimerization or DR4/DR5 receptor internalization - are also important. Therefore, HS levels do not play a sole determining role in TRAIL-induced apoptosis. Along the same line, the authors show that RPMI-8226 cell-surface HS promotes DR5 internalization despite the absence of direct DR5/heparin interactions. This suggests that HS at the cell surface may also affect apoptosis indirectly. To test this hypothesis, it would have been worthwhile to include the binding characteristics and HS-dependent internalization of DR4 into the study.

---

## [Referee Report · Reviewer #2 (Public Review)]

Summary:

In the manuscript by Luo et al, the authors investigated the nature and function of TRAIL-HS binding for the regulation of TRAIL-mediated apoptosis in cancer cells. The authors discovered that TRAIL binds to 12mer HS and identified the amino acid residues critical for the binding. The authors further nicely showed that 12mer HS binds to TRAIL homotrimer and larger HS can further promote the formation of larger TRAIL oligomers. Structural analyses were conducted to characterize the binding of TRAIL/HS complexes. At functional level, the authors demonstrated that HS promotes the cell surface binding of TRAIL to enhance TRAIL-mediated apoptosis in a variety of cancer cells. Moreover, the ability of TRAIL to induce apoptosis is correlated with cell surface HS level. Lastly, the authors showed that HS forms complex with TRAIL and its receptor DR5 and promotes DR5 internalization.

Strengths:

Overall, this is a nicely executed study providing both mechanistic and functional insight for TRAIL-mediated apoptosis. It conducted detailed characterization on the direct binding between HS and TRAIL and provided solid evidence supporting the role of such interaction for the regulation of TRAIL-induced apoptosis. The experiments were well-designed with proper controls included. The data interpretation is accurate. The manuscript was clearly written and easy to follow by general readers.

Weaknesses:

There is no major weakness identified from this study. As the authors pointed out, the current relationship between cell surface HS level and sensitivity to TRAIL-mediated apoptosis is still correlative and will need further investigation in the future.

---

## [Author Response]

The following is the authors’ response to the original reviews.

**Reviewer #1 (Recommendations For The Authors):**
This reviewer found the paper of very high interest, well supported, and well written. I have only a few suggestions to the authors for further improvement:1. TRAIL mutants carrying individual mutations of basic residues R119, R122 and K125 were tested, but a TRAIL mutant lacking all three residues was not. This combined mutant protein would have allowed to test whether all heparin binding is abolished (e.g. that no other residues contribute to HS binding) and could have also been used as an independent control replacing heparin and heparinase treatment in binding/apoptosis studies. Given that the DR4/5 and heparin binding sites of TRAIL do not overlap, this form would be useful in determining the extent to which HS contributes to, or serves as a prerequisite for TRAIL binding to its receptor and cell death. Moreover, if bound to the receptor, this mutant TRAIL is expected to completely prevent HS-mediated receptor internalization. The added value of this experiment therefore is that it may provide an answer to the controversial debate on whether DR receptor internalization promotes or inhibits apoptosis.In Fig. 5C, we provided data showing that the binding of R115A mutant of hTRAIL (equivalent to murine R199A mutant) to MB-453 cells was very similar to the binding of WT hTRAIL to heparin lyase treated cells. This finding suggests that nearly all HS-dependent binding to cell surface HS was abolished by mutating R115. Since a single mutant is sufficient, we felt there is little point in combining multiple mutations. We also used R115A mutant as an independent control replacing heparin and heparinase treatment in apoptosis assay in Fig. 7E. With regard to using the mutant in the internalization assay, we thank the reviewer for this excellent suggestion and will incorporate it into our future study as we intend to perform more in-depth investigation on the exact mechanism of internalization.1. The domain data is interesting, but its physiological significance remains obscure and it also somewhat distracts from the main theme of the study. It may be removed from a revised manuscript.

We partially agree with the reviewer’s assessment, but we felt that this discovery is of sufficient novelty and should be made known to the whole community.

1. TUNEL data is shown as a picture in Figure 6, but quantification is lacking.

We have included the statistics of the TUNEL data in the final version as Fig. 6D.

1. Is the HS20 antibody a well-suited pan-anti-HS antibody? Why was this antibody used instead of heparinase digestion followed by the use of HS "stub" antibodies that were previously used as a reliable readout for overall sulfation?

The HS20 mAb has been very well characterized by Dr. Mitchell Ho group (Gao et al., 2016). We have also done side-by-side comparison of HS20 and the most commonly used anti-HS mAb 10E4 by immunostaining and FACS. In nearly all tissues and cells tested, HS20 gave better sensitivity and lower background (after heparin lyase treatment) compared to 10E4. The staining pattern of the two mAbs are usually identical, but the signal/noise ratio of HS20 is much better than 10E4. The HS ”stub” antibody can be useful in certain applications, but it is used mainly as an indicator of the distribution/abundance of HSPGs, rather than a readout of overall sulfation.

1. The discussion should be stripped from expressions such as interestingly, curiously, unexpectedly, certainly, undoubtedly and the like to improve readability. The manuscript should be checked for typos (for example surface plasma resonance line 473, was served line 481).

We thank the reviewer for the suggestions and many of these expressions were removed in the final version.

1. Last but not least: to test the physiological relevance of these findings, it would be of the highest interest to use a mouse model harboring a tumor cell line of choice and derived lines with impaired or increased HS expression, as outlined in my public comments, and to test tumor responsiveness to TRAIL treatment. If already planned, I wish you Good Luck with the experiments!

We thank the reviewer for this excellent suggestion and we have indeed planned to do exactly that!

**Reviewer #2 (Recommendations For The Authors):**
1. The authors showed in Fig.2 that 12mer HS forms complex with TRAIL homotrimer. Please clarify if 12mer HS binding leads to the formation of the TRAIL homotrimer or TRAIL can form homotrimer in the absence of HS binding. Do the TRAIL mutations that affect HS binding, such as R115A, also impact the homotrimer formation?

TRAIL automatically forms a homotrimer independent of HS. It is known that formation of the homotrimer critically depends on a zinc ion, which is located on the threefold axis of the trimer and is bound by cysteine 240. We have also verified that all TRAIL mutants remain homotrimeric by size exclusion chromatography.

1. Does 12mer HS also suppress TRAIL-mediated apoptosis in MDA-MB-453 cells?

We thank the reviewer for this question but felt performing this experiment will not add any more insight to the main conclusion. Most likely, the result will be similar to what we saw in Fig. 7D, where we found 12mer significantly inhibits TRAIL-induced apoptosis, but inhibits less efficiently compared to heparin.

1. The authors nicely showed the correlation between surface HS level and sensitivity to TRAIL-induced apoptosis in MM cell lines and implicated that such correlation could be related with the difference in the expression level of SDC1. This is an interesting point worth further validation. Does ectopic SDC1 expression in IM-9 cells lead to increase cell surface HS and sensitivity to TRAIL treatment? On the other hand, will depletion of SDC1 expression in U266 or RPMI8226 cells decrease their sensitivity to TRAIL treatment?

We agree that this would be an excellent experiment to try and have actually attempted to overexpress SDC1 in IM-9 cells. But we found IM-9 cells are very difficult to transfect and we only managed to convert a small percentage of SDC1 negative cells to positive cells. Also, the level of SDC1 expression on the SDC1-positive cells was not changed after overexpression. We have not tried depleting SDC1 expression in U266 and RPMI8226 cells because such an experiment might change the property of these cells in unexpected ways, which would make result interpretation impossible. A previous report has shown that knocking down SDC1 could enhance clustering of TRAIL receptors in H929 cells (Wu et al., J Immunol 2012;), which actually led to slightly increased apoptosis.